

# Riming-dependent Snowfall Rate and Ice Water Content Retrievals for W-band cloud radar

Nina Maherndl[1], Alessandro Battaglia[2,3], Anton Kötsche[1], and Maximilian Maahn[1]

[1]Leipzig Institute of Meteorology (LIM), Leipzig University, Leipzig, Germany
[2]Politecnico of Torino, Torino, Italy
[3]University of Leicester, Leicester, UK

**Correspondence:** Maximilian Maahn (maximilian.maahn@uni-leipzig.de)

**Abstract.**

Accurate measurements of snowfall in mid- and high-latitudes are particularly important, because snow provides a vital freshwater source, and impacts glacier mass balances as well as surface albedo. However, ice water content (IWC) and snowfall rates (SR) are hard to measure due to their high spatial variability and the remoteness of polar regions. In this study, we present novel ice water content - equivalent radar reflectivity (IWC-$Z_e$) and snowfall rate - equivalent radar reflectivity (SR-$Z_e$) relations for 40° slanted and vertically pointing W-band radar. The relations are derived from joint in situ snowfall and remote sensing (W-band radar and radiometer) data from the SAIL site (Colorado, USA) and validated for sites in Hyytiälä (Finland), Ny-Ålesund (Svalbard), and Eriswil (Switzerland). In addition, gauge measurements from SAIL and Hyytiälä are used as an independent reference for validation. We show the dependence of IWC-$Z_e$ and SR-$Z_e$ on riming, which we utilize to reduce the spread in the IWC-$Z_e$ and SR-$Z_e$ spaces. Normalized root mean square errors (NRMSE) are below 25% for IWC > $0.1\,\mathrm{g\,m^{-3}}$. For SR, the NRMSE is below 70% over the whole SR range. We also present relations using liquid water path as a proxy for the occurrence of riming, which can be applied to both ground-based and space-borne radar-radiometer instruments. The latter is demonstrated using the example of the proposed ESA Earth Explorer 11 candidate mission WIVERN. With this approach, NRMSE are below 75% for IWC > $0.1\,\mathrm{g\,m^{-3}}$ and below 80% for SR > $0.2\,\mathrm{mm\,hr^{-1}}$.

## 1 Introduction

At mid- and high-latitudes, most precipitation stems from ice clouds (Mülmenstädt et al., 2015). Solid precipitation in the form of snow plays an important role in the Earth's hydrological cycle and energy budget, affecting surface albedo, glacier mass balance, freshwater storage, and cloud lifetime. However, ice water content (IWC) and snowfall rates (SR) are difficult to measure in part due to their high spatial variability. At high latitudes, ground-based precipitation observations are sparse and complicated by harsh environmental conditions (e.g., Førland et al., 2011).

Precipitation gauges are commonly used to measure liquid equivalent SR amounts. While gauges provide direct measurements of SR, they are prone to large uncertainties (e.g., Saltikoff et al., 2015). Particle size and velocity size distribution data from snowfall cameras can also be used to estimate SR, given the observational volume is large enough. The advantage of this approach over gauge measurements is that IWC and SR can jointly be derived with a high temporal resolution. However, the





particle mass distribution cannot be measured directly with optical instruments, thus mass-size relations need to be assumed
from literature (e.g., Heymsfield et al., 2016) or retrieved (e.g., von Lerber et al., 2017). SR derived from radar can provide
more information on the spatial distribution as compared to the point-measurement of a gauge or snowfall camera. In addition,
radar observations are vertically resolved and can be used to derive vertical profiles of IWC. W-band radars, which operate
at about 94 GHz, are commonly used due to their high sensitivity to cloud particles. Space-borne W-band radar can provide
global observations of IWC and SR as demonstrated by the CloudSat Cloud Profiling Radar (Tanelli et al., 2008), that has
provided the first global climatology of SR (Hiley et al., 2011; Milani et al., 2018) and, combined with the CALIPSO lidar,
of IWC (e.g., Delanoë and Hogan, 2010). However, current satellite-derived snowfalls products suffer from poor sampling
(Scarsi et al., 2024) and a "blind-zone" close to ground, thus missing shallow precipitation (Maahn et al., 2014; Schirmacher
et al., 2023). Further, the information content of satellite observations is typically not sufficient to constrain the highly variable
microphysical properties of snow and ice particles unambiguously.

SR and IWC cannot be measured directly by radar, but has to be inferred from radar reflectivity. Power law relations between
$z_e$ in linear units, defined as $z_e[\mathrm{mm^6 m^{-3}}] = 10^{0.1 \cdot Z_e[\mathrm{dBZ}]}$, and IWC or $z_e$ and SR in the form of $z_e = a \cdot \mathrm{IWC}^b$ and $z_e = c \cdot \mathrm{SR}^d$
are commonly used for IWC and SR estimation (e.g., Fuller et al., 2023, provide and overview of SR-$Z_e$ relations for W-band
radar). These relations show differences of about one order of magnitude in estimates of IWC and SR. The large spread stems
from the large variability among ice and snow particle distributions (PSDs), density, shape, orientation, crystal habit, etc.
Although these relations can have significant uncertainties for individual cases, they are successfully applied to space-borne
radar data sets because the random errors cancel partly out in seasonal time scales (Kulie and Bennartz, 2009).

To reduce the variability in $Z_e$-IWC and $Z_e$-SR space, additional variables are commonly included in retrievals like air
temperature $T$ for both ground-based and space-borne radar (e.g., Wood and L'Ecuyer (2021)) or polarimetric variables such
as KDP or ZDR for ground-based radar (Bukovčić et al. (2020), for S-band radar). Recent studies have demonstrated the
potential of including brightness temperature $T_B$ and/or nadir Doppler observations (like available for the EarthCARE radar,
Illingworth et al., 2015; Kollias et al., 2023) to constrain SR estimates from space (Battaglia and Panegrossi, 2020; Mroz et al.,
2023). $T_B$ and/or Doppler can give insight on the location and amount of supercooled liquid water layers, which can lead to
higher ice particle densities due to supercooled droplets freezing onto ice particles upon contact, which is commonly referred
to as riming. Riming drives $Z_e$ variability (Maherndl et al., 2024b) and Fuller et al. (2023) show that most literature SR-$Z_e$ lead
to strong biases when applied to rimed snow particles. Fuller et al. (2023) argue new research is needed to refine the SR–$Z_e$
relationship for rimed snow particles.

WIVERN (WInd VElocity Radar Nephoscope, Illingworth et al., 2018; Battaglia et al., 2022), one of the two remaining
ESA Earth Explorer 11 candidate missions, is planned to be equipped with a conical scanning 94 GHz radar and a passive 94
GHz radiometer. While the main objective of the mission is to measure global in-cloud winds (e.g., inside tropical cyclones,
Tridon et al., 2023), WIVERN reflectivity data can also be used to derive IWC and SR. Compared to CloudSat (Tanelli
et al., 2008) and EarthCARE (Illingworth et al., 2015), WIVERN's 800 km swath provides better coverage (70 times better
than CloudSat) leading to significantly reduced the uncertainty of polar snowfall estimates (Scarsi et al., 2024). In addition,





WIVERN's 42° angle of incidence results in a thinner radar blind zone near the surface (especially over the ocean) (Coppola
et al., 2024).

In this study, we present $Z_e$-IWC and $Z_e$-SR relations for both 40° slanted and vertically pointing W-band radar. The relations were derived from joint radar and in situ snowfall observations during winter 2022/2023 in Gothic (Colorado, USA) and validated for additional mid- and high-latitude sites in Hyytälä (Finland), Ny-Ålesund (Svalbard Norway), and Eriswil (Switzerland). We investigate the dependence of $Z_e$-IWC and $Z_e$-SR on snow particle riming based on joint in situ and radar data. Further, we include liquid water path (LWP) as an additional parameter in the relations as a proxy for the occurrence of riming (Moisseev et al., 2017). This approach allows to reduce uncertainty in the $Z_e$-IWC and $Z_e$-SR spaces when in situ data is not available. The novel relations can therefore be applied to both ground-based and space-borne radar (and radiometer). The latter is demonstrated with synthetic WIVERN observations accounting for the space-borne geometry and estimated uncertainties.

The paper is structured as follows. We first give an overview of all ground-based measurement sites and instruments we use to derive our reference data in Sect. 2. In Sect. 3, we 1. explain the riming retrieval we use, 2. demonstrate a novel approach allowing us to correct $Z_e$ for the 40° viewing angle, 3. describe the reference IWC and SR data, and 4. show our methods to fit $Z_e$-IWC and $Z_e$-SR relations. Results of the fits are presented in Sect. 4.1, and their performance is evaluated in Sect. 4.2. Sect. 5 draws a conclusion.

## 2 Measurement Sites and Instruments

We use data from the the U. S. Department of Energy Atmospheric Radiation Measurement (ARM) user facility's Surface Atmosphere Integrated Field Laboratory (SAIL, Feldman et al., 2023) site in Gothic, Colorado (USA) to develop $Z_e$-SR and $Z_e$-IWC relations for (slanted) W-band radar. The performance of the new relations are tested using data from three additional mid- and high-latitude sites at Hyytiälä (Finland), Ny-Ålesund (Svalbard Norway), and Eriswil (Switzerland). In the following, we describe the measurement sites and the main instrumentation used in this study.

### 2.1 Field Experiment at the SAIL Site

In winter 2022/23, the Leipzig University 94 GHz radar (LIMRAD94) and a Video In Situ Snowfall Sensor (VISSS, Maahn et al., 2024) were deployed at the SAIL site in Gothic, Colorado, USA, (38.95621°N, 106.98796°W; 2885 m above mean sea level (MSL), Feldman et al., 2023). LIMRAD94 is a polarimetric simultaneous transmission simultaneous reception (STSR) Doppler cloud radar manufactured by Radiometer Physics GmbH (RPG, instrument type RPG-FMCW-94-DP, Küchler et al., 2017). The radar scanning strategy included slanted observations at a constant angle of 40° during December 2022 and January 2023, and range height indicator (RHI) scans in February 2023 (Kalesse-Los et al., 2023). LIMRAD94 (at 2905 m MSL) was operated at a range resolution of about 12 m for ranges below 2000 m, which corresponds to a vertical resolution of 7.7 m below 1288 m at the 40° observation angle. The VISSS (at 2885 m MSL) was deployed below the line of sight of the radar, at a horizontal distance of about 500 m. The VISSS consists of two cameras with telecentric lenses, mounted perpendicular to each





other. The set-up allows for accurate characterization of snow particle shape and size. At the SAIL site, the first generation VISSS, here denoted VISSS1, was deployed. VISSS1 has a pixel resolution of $58.832\,\mu\mathrm{m}\,\mathrm{px}^{-1}$, a frame rate of 140 Hz, and an observation volume of $wxdxh$ = 75.2 x 75.2 x 60.1 $\mathrm{mm}^3$. VISSS data products relevant to this study include time averaged particle size distributions (PSDs) and sedimentation velocity distributions.

We use additional data acquired by ARM of near-surface air temperature $T$, SR from a Pluvio weighing precipitation gauge and liquid water path (LWP). The latter product is derived from a site-specific statistical retrieval from microwave radiometer brightness temperature measurements. LWP, $T$, and Pluvio SR data are obtained from the ARM data portal (https://adc.arm.gov/discovery/, last access: 18 Nov 2024).

## 2.2    Additional mid- and high-latitude sites for validation

For validation and evaluation we use joint vertically-pointing 94 GHz radar and VISSS observations obtained at the Hyytiälä Forestry Field Station of the University of Helsinki, Finland (HYY), in 2021/21 and 2023/24, at Eriswil, Switzerland, during the PolarCAP (Polarimetric Radar Signatures of Ice Formation Pathways from Controlled Aerosol Perturbations) campaign 2023/24, and at the French German atmospheric observatory AWIPEV (named after the Alfred Wegener Institute for Polar and

Marine Research and the French Polar Institute Paul Emile Victor, Ebell et al., 2020) in Ny-Ålesund (NYA), Svalbard, from 2021 onward (Fig. 1). The selected time periods used in this study are the result of several criteria: availability of joint in situ and radar data, snowfall at the ground, and temperatures below -1° C to remove melting snowflakes, which are not represented in our scattering simulations.

### 2.2.1    Measurement Site Hyytiälä

In Hyytiälä, Finland, the University of Helsinki operates a Forestry Field Station (61.84398°N, 24.28758°E; 150 m MSL). The station is equipped with a vertically pointing, 94 GHz cloud radar by RPG (instrument type RPG-FMCW-94-DP). The radar has a range resolution of about 25.5 m in the height range near-ground that we use. VISSS1 was deployed in the field close to the radar during winter 2021/2022. Since November 2023, the third generation VISSS (VISSS3) is set up there which has a pixel resolution of $46.0\,\mu\mathrm{m}\,\mathrm{px}^{-1}$, a frame rate of 220 Hz, and an observation volume of $wxdxh$ = 47.1 x 47.1 x 58.9 $\mathrm{mm}^3$.

Near-surface air temperature $T$ from the site's weather station and the LWP product from a HATPRO microwave radiometer (Rose et al., 2005) are used as auxiliary data. Equivalent liquid SR data from a Pluvio gauge is used for validation. Radar, LWP, Pluvio SR, and $T$, data are accessed via the Cloudnet data portal (Moisseev and Petäjä, 2024).

### 2.2.2    Measurement Site Ny-Ålesund

The joint French-German Arctic research station AWIPEV is located in Ny-Ålesund, Svalbard (78.92308°N, 11.92108°E;

11 m MSL). On the roof of the AWIPEV observatory a 94 GHz radar is operated by the University of Cologne (JOYRAD-94). JOYRAD-94 is a non-scanning, Doppler cloud radar manufactured by RPG (instrument type RPG-FMCW-94-SP). JOYRAD-



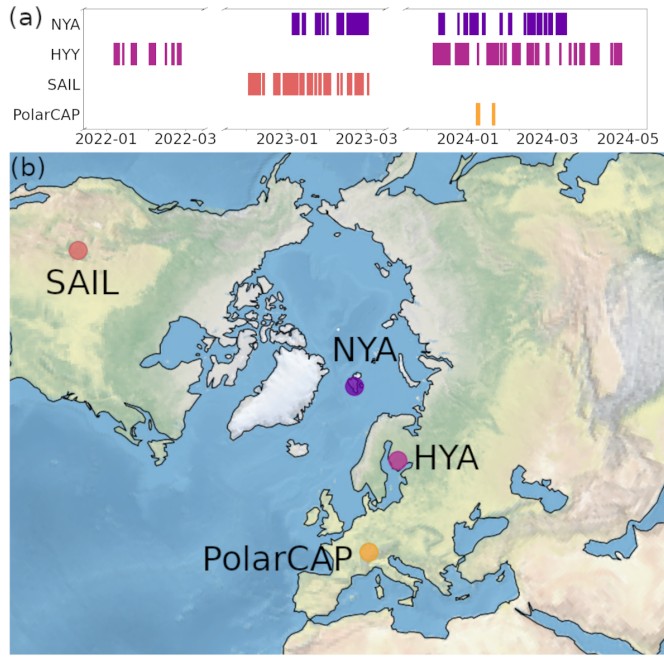

**Figure 1.** (a) temporal coverage and (b) locations of evaluation datasets: Ny-Ålesund site (NYA) in Svalbard; Hyytiälä site (HYY) in Finland; SAIL site in Crested Butte, Colorado, USA; PolarCAP campaign in Eriswil, Switzerland. For SAIL, polarimetric W-band measurements at 40° elevation were obtained.

94 has a range resolution of about $3.2\,\mathrm{m}$ in the height range we are interested in. Since December 2021, the second generation VISSS (VISSS2) is located on the measurement field close to the observatory. VISSS2 has a pixel resolution of $43.266\,\mathrm{\mu m\,px^{-1}}$, a frame rate of $250\,\mathrm{Hz}$, and an observation volume of $wxdxh$ = 55.2 x 55.2 x 44.2 $\mathrm{mm^3}$. Additionally, we use near-surface air

temperature $T$ from the site and the LWP product from a HATPRO. JOYRAD-94, LWP, and $T$, data are accessed via Cloudnet (Ebell and Ritter, 2024).

### 2.2.3 Field Experiment at Eriswil, Switzerland

Similar to SAIL, LIMARD94 and VISSS1 were deployed jointly in Eriswil, Switzerland (47.07056°N, 7.87278°E; $921\,\mathrm{m}$ MSL) during the PolarCAP (Polarimetric Radar Signatures of Ice Formation Pathways from Controlled Aerosol Perturbations)

field experiment in winter 2023/24. The field experiment was conducted under the umbrella of the ERC research project CLOUDLAB (Henneberger et al., 2023) by ETH Zurich. LIMRAD94 was operated with a range resolution of about $12\,\mathrm{m}$ below $2000\,\mathrm{m}$. Auxiliary $T$, and LWP data (derived from a HATRPO) is available from the mobile exploratory platform LACROS of the Leibniz Institute for Tropospheric Research (TROPOS) and accessed via Cloudnet (Seifert, 2024). Only a small subset of the campaign data can be used for this study due to warm near-surface temperatures at Eriswil during PolarCAP.





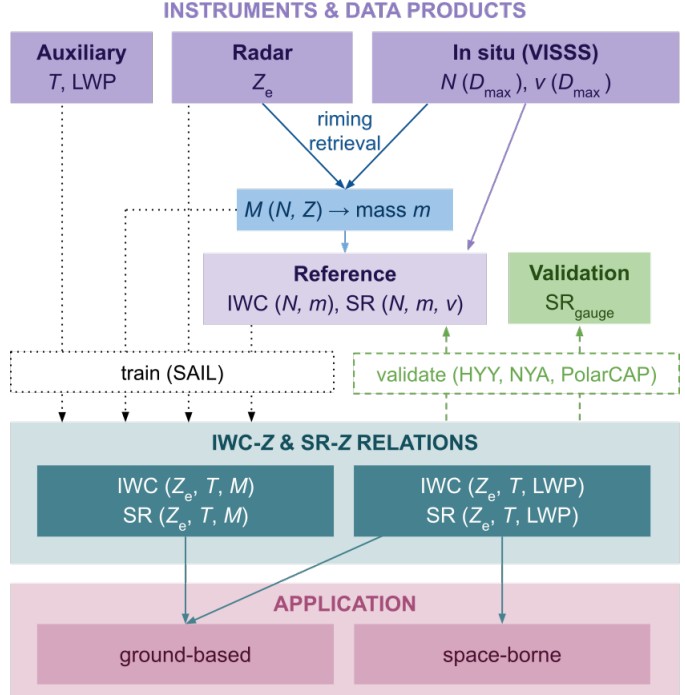

**Figure 2.** Overall logic of the paper. The purple section gives an overview of the used instruments and data products, where $Z_e$ is the W-band radar reflectivity, $D_{\max}$ the maximum dimension of snow and ice particles, $N$ the number concentration, $v$ the fall velocity, $M$ the normalized rime mass, $m$ the particle mass, $T$ the air temperature, LWP the liquid water path, SR the snowfall rate and IWC the ice water content. The data products from the SAIL site are used to train the IWC and SR relations shown in the mint section. Data from the HYY, NYA, and PolarCAP sites are used for validation (green). The relations depending on $M$ can be applied to ground-based data; the relations depending on LWP can be applied to both ground-based and space-borne data (pink section).

## 3 Methods

Figure 2 summarizes the overall logic of the paper and gives an overview of the methods described in the following sections. We use ground-based in situ and radar data to derive reference IWC and SR (Sect. 3.2). A riming retrieval is used to get more accurate estimations of snow particle masses (Sect. 3.1). In addition, auxiliary data is used for the retrieval development and validation. Two variants of IWC and SR relations (Sect. 3.4) are derived based on SAIL data and validated with HYY, NYA, and PolarCAP data: 1. depending on radar reflectivity $Z_e$, air temperature $T$ and normalized rime mass $M$ and 2. depending on radar reflectivity $Z_e$, air temperature $T$ and LWP. We show the applicability of 1. to ground-based and of 2. to both ground-based and space-borne data.



### 3.1 Normalized rime mass retrieval

We take advantage of the joint radar and in situ observations to quantify ice and snow particle riming. To describe riming, we
use the normalized rime mass $M$ introduced by Seifert et al. (2019). $M$ is defined as the particle's rime mass $m_{\mathrm{rime}}$ divided by
the mass of a size-equivalent spherical graupel particle $m_{\mathrm{g}}$, where we assume a rime density of $\rho_{\mathrm{rime}} = 700 \mathrm{\ kg \ m^{-3}}$:

$$M = \frac{m_{\mathrm{rime}}}{m_{\mathrm{g}}}, \tag{1}$$

where

$$m_g = \frac{\pi}{6} \rho_{\mathrm{rime}} D_{\mathrm{max}}^3. \tag{2}$$

$D_{\mathrm{max}}$ is the maximum dimension defined as the diameter of the smallest circle encompassing the ice particle in $\mathrm{m}$.

We use the combined method from Maherndl et al. (2024a) to retrieve $M$, which was originally developed for airborne data.
Here, we adopted the method for application to ground-based data. In the following, we give a brief description of the retrieval
and our adaptions for ground-based data. For more details, we refer the reader to Maherndl et al. (2024a).

The combined method derives a time series of $M$ from collocated PSD and radar reflectivity $Z_{\mathrm{e}}$ measurements. Here, we
assume PSDs derived from VISSS observations at the ground are representative of particles in the minimum radar measurement
volume above ground. For SAIL, we use the radar range gate in an altitude of about 355 m, which is located closest to VISSS
due to the radar elevation angle. We derive the standard deviation of $Z_{\mathrm{e}}$ between 410 m and 355 m (corresponding to five
range gates) and remove all time stamps with standard deviation larger 2 dB. This is done to remove times with strong vertical
gradients of $Z_{\mathrm{e}}$ close to ground, where the assumption that the PSD does not change from the radar range gate to the ground
does not hold. Further, we filter for $Z_{\mathrm{e}} > -5 \mathrm{\ dBZ}$ to remove very light snowfall cases. For HYY, NYA, and PolarCAP, we select
the closest range gate to the ground, i.e., the range gate above the minimum measurement range (corresponding to altitudes of
about 100-150 m) and derive standard deviations of $Z_{\mathrm{e}}$ over all range gates below 200 m. We also filter for standard deviations
smaller than 2 dB and $Z > -5 \mathrm{\ dBZ}$. PSDs and $Z_{\mathrm{e}}$ are averaged for 100 s to account for the different observational volume
(at least to a certain extent). We tested different time offsets of up to 5 minutes between radar and VISSS to account for the
typical sedimentation time of snow particles to the ground. However, we found that the $M$ results did not change within the
retrieval uncertainties and therefore chose to use no time offset. 100 s averaging windows corresponds to a spatial distance of
about 1 km assuming a horizontal wind speed of $10 \mathrm{\ m s^{-1}}$.

The retrieval uses Optimal Estimation (Rodgers, 2000) with the pyOptimalEstimation Python library (Maahn et al., 2020)
to derive $M$ by forward simulating $Z_{\mathrm{e}}$ based on the observed in situ PSD and and comparing to the matched, observed $Z_{\mathrm{e}}$.
As forward operator, the Passive and Active Microwave radiative TRAnsfer tool (PAMTRA, Mech et al., 2020) is used,
which includes empirical relationships from Maherndl et al. (2023) for estimating particle scattering properties based on the
Self-Similar Rayleigh-Gans Approximation (SSRGA, Hogan and Westbrook, 2014; Hogan et al., 2017) as a function of $M$.
Maherndl et al. (2023) assumed horizontally aligned ice particles viewed by vertically pointing radar. We therefore recalculated
the SSRGA coefficients for a viewing angle of 40° to be applicable for the slanted SAIL data and present the results in Appendix





A. Particle mass $m(D_{\max})$ is approximated by a power law relation with prefactor $a_m$ and exponent $b_m$

$$m(D_{\max}) = a_m D_{\max}^{b_m}. \tag{3}$$

We use the riming-dependent mass-size parameters $a_m$ and $b_m$ (i.e., the "mean" parameters from Maherndl et al., 2023) that were estimated for different degrees of riming, i.e., $M$ values. In Maherndl et al. (2023), discrete mass-size parameter are given, which we interpolate for continuous $M$. Because currently no particle classification product is available for all sites and

mass-size parameter variability is rather dominated by riming than by particle shape, we assume a mixture of particle shapes (columns, dendrites, needles, plates, rosettes) and use the "mean" mass-size parameters, which are closest to the parameters for aggregates of plates. Maherndl et al. (2024a) investigated the dependence of the retrieved $M$ on the particle shape assumption and showed that assuming plates or dendrites result in the same $M$ within the retrieval uncertainty estimates. $M$ results assuming columns are slightly lower than assuming dendrites. Our results could therefore have a slight positive bias during

snowfall events with column-like shapes.

The $M$ retrieval results are used for multiple purposes in this study. First, we use $M$ to estimate particle masses by choosing the appropriate parameters from Maherndl et al. (2023) for each time step (Sect. 3.2). Second, we use $M$ to select time periods with predominately unrimed particles to derive a relation between $Z_{\mathrm{e}}$(IWC) for vertically pointing radar and $Z_{\mathrm{e}}$(IWC) for a viewing angle of 40° (Sect. 3.3). Third, we investigate the dependence of $Z_{\mathrm{e}}$-IWC and $Z_{\mathrm{e}}$-SR relation on $M$ (Sect. 4.1.1).

**3.2 Reference IWC and SR data**

To derive IWC from in situ PSD observations, size-resolved ice particle mass must be assumed. For our IWC reference dataset, IWC is calculated by summing the product of ice particle mass $m(D_{\max})$ and VISSS observed $N(D_{\max})$ for the lower to upper size ranges of the VISSS, $D_{\mathrm{lower}}$ to $D_{\mathrm{upper}}$

$$\mathrm{IWC} = \sum_{D_{\mathrm{lower}}}^{D_{\mathrm{upper}}} m(D_{\max})\, N(D_{\max})\, \Delta D_{\max}, \tag{4}$$

where $\Delta D_{\max}$ is the size bin width. $N(D_{\max})$ is taken from the "level2match" VISSS data, where particles observed with both VISSS cameras are matched and binned particle properties are available as a function of time either from one of the cameras or using the minimum, average, or maximum from both cameras. We use the maximum $D_{\max}$ observed from both cameras for each matched particle to approximate the true $D_{\max}$. $m(D_{\max})$ is approximated by a power law relation with $M$-dependent mass-size parameters as described in Sect. 3.1.

SR is calculated by summing the product of ice particle mass $m(D_{\max})$, VISSS observed $N(D_{\max})$, and VISSS observed particle sedimentation speed $v(D_{\max})$ for the lower to upper size ranges of the VISSS, $D_{\mathrm{lower}}$ to $D_{\mathrm{upper}}$

$$\mathrm{SR} = \sum_{D_{\mathrm{lower}}}^{D_{\mathrm{upper}}} m(D_{\max})\, N(D_{\max})\, v(D_{\max})\Delta D_{\max}. \tag{5}$$

Because sedimentation velocity can only be determined for a subset of observed particles, who are detected multiple times (see Maahn et al., 2024) NaN values must be interpolated. To avoid unrealistic behavior at the edges of the size spectrum, NaN

values of $v$ are filled with $v$ from the closest available size bin.



### 3.3 Viewing angle correction

Falling ice and snow particles typically orient themselves horizontally in the atmosphere (List and Schemenauer, 1971; Zik-munda and Vali, 1972; Wang, 2021; Stout et al., 2024), thus their radar reflectivity depends on the viewing angle. Because only vertically-pointing radar observations are available for the validation sites, a $Z_e$ correction must be applied to compare to the

40° observation angle at SAIL. To derive the correction term, HYY, NYA, PolarCAP, and SAIL data are filtered for $M < 0.01$ to get all time intervals with (predominately) unrimed particles. Because for PolarCAP only 155 data points remain, PolarCAP data is excluded in the further steps. Then, median $Z_e$ for IWC in 30 logarithmic bins between $10^{-5}\,\mathrm{kg\,m^{-3}}$ and $10^{-2}\,\mathrm{kg\,m^{-3}}$ are derived for HYY, NYA, and SAIL. Logarithmic bins were chosen because the reference IWC data follows approximately a normal distribution in logarithmic space; the number of bins was selected such that there is a sufficient amount of data points

per bin.

The results show (Fig. 3) that medians for HYY and NYA are nearly identical and therefore a joint median is derived. The reduction of median $Z_e$ by using slanted observations at SAIL instead of vertically pointing observations at HYY and NYA between the vertically pointing is nearly constant with IWC and results to $2.29 \pm 0.39\,\mathrm{dB}$ (mean $\pm$ standard deviation). Thus, the offset can be subtracted from the vertically-pointing $Z_e$ data to correct for the 40° observation angle. To test whether

radar calibration or climatological differences causes the derived offset instead of the viewing angle, we performed a similar analysis comparing 90° SAIL observations for the time when they when available together with the 40° data in February 2023. The threshold for unrimed particles had to be increased to $M < 0.02$, to have a sufficient number of data per IWC bin for the analysis ($M < 0.02$: on average over 30 per bin and 700 data points in total; $M < 0.01$: only 200 data points in total). We found a similar offset of $2.25 \pm 0.80\,\mathrm{dB}$, albeit with a higher standard deviation likely due to the smaller number of observations. This

indicates the offset is indeed caused by viewing angle. Distributions of 40° and (corrected) 90° $Z_e$ during scans in Feb 2023 are shown in Appendix B. The offset likely depends on properties such as the PSD in addition to particle orientation. We tested the dependence on particle riming by performing a similar analysis for specific $M$ ranges. We found the same offset within the respective standard-deviation ranges, albeit with larger standard deviations for larger $M$ values, likely due to the smaller amount of data. We hypothesize that for single events the offset might differ but averages to the derived value over longer time

spans for the analyzed sites.

### 3.4 Deriving IWC-$Z_e$ and SR-$Z_e$ relations

The reference IWC in $\mathrm{kg\,m^{-3}}$ and SR in liquid water equivalent $\mathrm{mm\,hr^{-1}}$ (Sect. 3.2) are related to the equivalent radar reflec-tivity factor close to ground $z_e$ in linear units $\mathrm{mm^6\,m^{-3}}$, near-surface air temperature $T$ in °C, and normalized rime mass $M$ for $M > 0$:

$$\mathrm{IWC}\,[\mathrm{kg\,m^{-3}}] = p_1 \cdot z_e^{p_2} \cdot 10^{p_3 \cdot T} \cdot M^{p_4}, \tag{6}$$

and

$$\mathrm{SR}\,[\mathrm{mm\,hr^{-1}}] = p_5 \cdot z_e^{p_6} \cdot 10^{p_7 \cdot T} \cdot M^{p_8}, \tag{7}$$





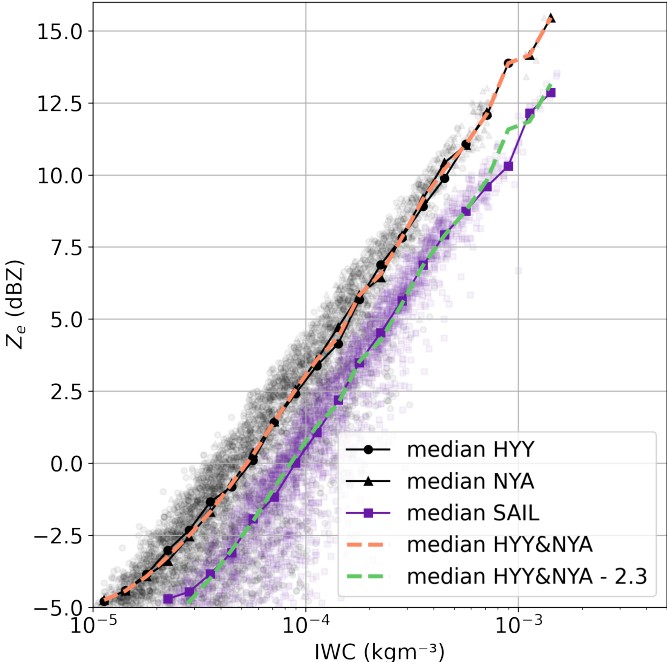

**Figure 3.** W-band $Z_e$ as a function of IWC derived for unrimed particles for vertically pointing radar at Hyytiälä (HYY), Finland, and Ny-Ålesund (NYA), Svalbard (grey points) and for 40° observations from the SAIL site in Gothic, Colorado, USA (violet points). Data suggest an offset correction of $2.25 \pm 0.80$ dB. See text for further explanations.

where $p_i$ are the respective fit coefficients. $z_e$ in linear units is converted to $Z_e$ in logarithmic units with

$Z_e[dBZ] = 10 \cdot \log_{10}(z_e[\mathrm{mm^6 m^{-3}}])$. A multi-linear regression is performed to derive the coefficients $p_i$, which are presented in Sect. 4.1.1.

However, $M$ is typically not available for sites without in situ PSD data. Therefore, we also relate the reference IWC in $\mathrm{kgm^{-3}}$ and SR in liquid water equivalent $\mathrm{mmhr^{-1}}$ to $z_e$ in $\mathrm{mm^6 m^{-3}}$, $T$ in °C, and LWP in $\mathrm{kgm^{-2}}$, which should indicate periods, where riming is likely (Moisseev et al., 2017):

$$\mathrm{IWC} \, [\mathrm{kgm^{-3}}] = q_1 \cdot z_e^{q_2} \cdot 10^{q_3 \cdot T} \mathrm{LWP}^{q_4}, \tag{8}$$

and

$$\mathrm{SR} \, [\mathrm{mmhr^{-1}}] = q_5 \cdot z_e^{q_6} \cdot 10^{q_7 \cdot T} \cdot \mathrm{LWP}^{q_8}, \tag{9}$$

where $q_i$ are the respective fit coefficients. Again, a multi-linear regression is performed to derive the coefficients $p_i$, which are presented in Sect. 4.1.2.





**Table 1.** IWC in $\mathrm{kgm^{-3}}$ and SR in $\mathrm{mmhr^{-1}}$ fit coefficients using $z_\mathrm{e}$ in $\mathrm{mm^6m^{-3}}$, $T$ in °C, and $M$.

| IWC coefficients | | | | SR coefficients | | | |
|---|---|---|---|---|---|---|---|
| $p_1$ | $p_2$ | $p_3$ | $p_4$ | $p_5$ | $p_6$ | $p_7$ | $p_8$ |
| $1.17 \cdot 10^{-5}$ | 0.95 | -0.015 | -0.38 | 0.044 | 1.10 | 0.00053 | -0.31 |

$p_1$ is given in $\mathrm{kgm^{-3}}$ and $p_5$ in $\mathrm{mmhr^{-1}}$.

## 4 Results and Discussion

In this section, we first present our novel IWC-$Z_\mathrm{e}$ and SR-$Z_\mathrm{e}$ relations (Sect. 4.1). We show results for the respective fit coefficients using $Z_\mathrm{e}$, $T$, and $M$ (Sect. 4.1.1), and $Z_\mathrm{e}$, $T$, and LWP (Sect. 4.1.2). The latter can be applied when there is no in situ snowfall data, but a radiometer LWP product available, as is common for Cloudnet sites or certain space-borne instruments such as WIVERN. All relations are then evaluated against the reference IWC and SR dataset (described in Sect. 3.2) and their application to space-borne radar is tested using WIVERN as an example in Sect. 4.2. In addition, we compare the performance 255 of the IWC-$Z_\mathrm{e}$ relations to literature relations and evaluate results for the SR-$Z_\mathrm{e}$ relations with gauge data.

### 4.1 Empirical relations to derive IWC and SR

In general, the fit functions presented in the following should be applied to attenuation corrected 40° slanted $Z_\mathrm{e}$. By applying the correction term from Sect. 3.3, vertically pointing $Z_\mathrm{e}$ can also be used. Here, we only use $Z_\mathrm{e}$ data from ground-based radar close to ground (to be able to compare to in situ snowfall observations at ground). Attenuation due to atmospheric gases and 260 hydrometers from the ground to the near-surface radar volume can be neglected thus we did not perform attenuation corrections of $Z_\mathrm{e}$.

#### 4.1.1 Dependence on $Z_\mathrm{e}$, $T$, and $M$

Table 1 presents the fit coefficient results for Eq. 6 and Eq. 7. The resulting IWC-$Z_\mathrm{e}$ and SR-$Z_\mathrm{e}$ relations are shown together with the reference IWC and SR data in Fig. 4 for varying $M$ from unrimed ($M$<0.01) to spherical graupel ($M$=1.0). The 265 reference data set contains only few data points with $M$ close to 1.0, due to the rare occurrence of particle populations consisting only of dense, spherical graupel. IWC and SR for unrimed particles are generally higher at constant W-band $Z_\mathrm{e}$ than for rimed particles and decrease with increasing amounts of riming. The spread in IWC-$Z_\mathrm{e}$ or SR-$Z_\mathrm{e}$ space due to riming is stronger for IWC than SR. This is likely due to increased fall velocities of rimed particles, which result in higher SR, counteracting the IWC-$Z_\mathrm{e}$ spread.

#### 4.1.2 As functions of $Z_\mathrm{e}$, $T$, and LWP

Table 2 presents the fit coefficient results for Eq. 8 and Eq. 9. Figure 5 shows the resulting IWC-$Z_\mathrm{e}$ and SR-$Z_\mathrm{e}$ relations for varying LWP and $T$ conditions. Literature IWC-$Z_\mathrm{e}$ and SR-$Z_\mathrm{e}$ relations for W-band from Hogan et al. (2006); Protat





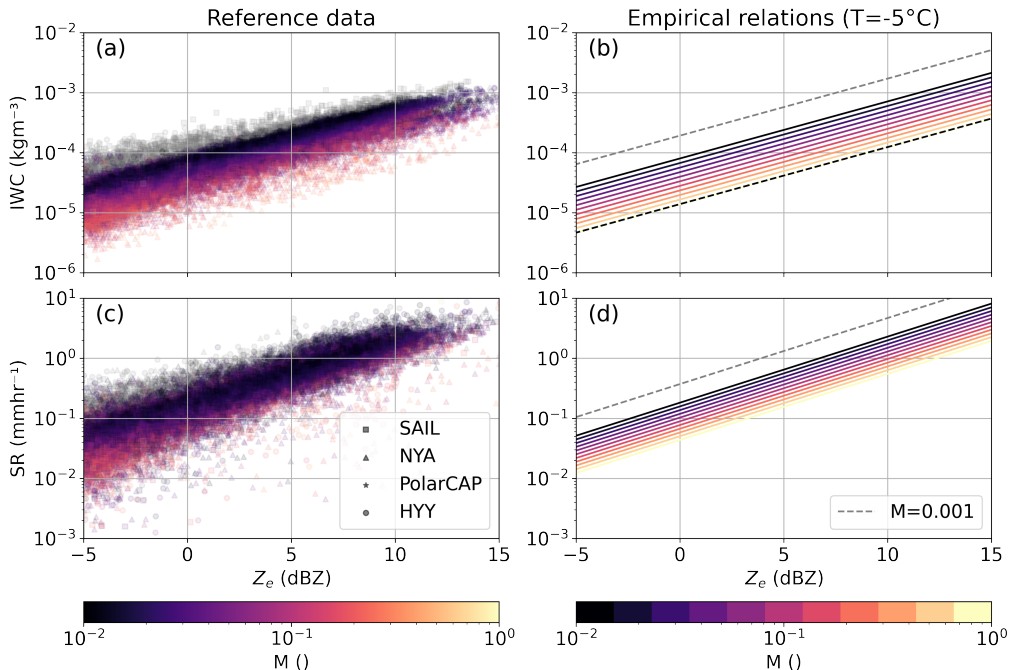

**Figure 4.** Reference data for (a) IWC-$Z_e$ and (b) SR-$Z_e$ for the different sites denoted with different symbols. $M$ is color coded; Data points with $M <0.01$ are considered (predominately) unrimed and shown in gray. (c) IWC-$Z_e$ and (d) SR-$Z_e$ empirical functions for $T$=-5°C and $M$ ranging from 0.01 (nearly unrimed) to 1.0 (spherical graupel). The respective function for $M$=0.001, which corresponds to the lowest 1% of $M$ retrieval results, is shown as a gray dashed line in (c) and (d).

**Table 2.** IWC in $\mathrm{kgm}^{-3}$ and SR in $\mathrm{mmhr}^{-1}$ fit coefficients using $z_e$ in $\mathrm{mm}^6\mathrm{m}^{-3}$, $T$ in °C, and LWP in $\mathrm{kgm}^{-2}$.

| | IWC coefficients | | | | SR coefficients | | | |
|---|---|---|---|---|---|---|---|---|
| LWP ($\mathrm{kgm}^{-2}$) | $q_1$ | $q_2$ | $q_3$ | $q_4$ | $q_5$ | $q_6$ | $q_7$ | $q_8$ |
| $\geq$0.1 | $1.93\cdot 10^{-5}$ | 0.94 | -0.045 | -0.23 | 0.096 | 1.05 | -0.020 | -0.13 |
| <0.1 | $4.39\cdot 10^{-5}$ | 1.01 | -0.016 | 0.0 | 0.13 | 1.16 | -0.0043 | 0.0 |

$q_1$ is given in $\mathrm{kgm}^{-3}$ and $q_5$ in $\mathrm{mmhr}^{-1}$.





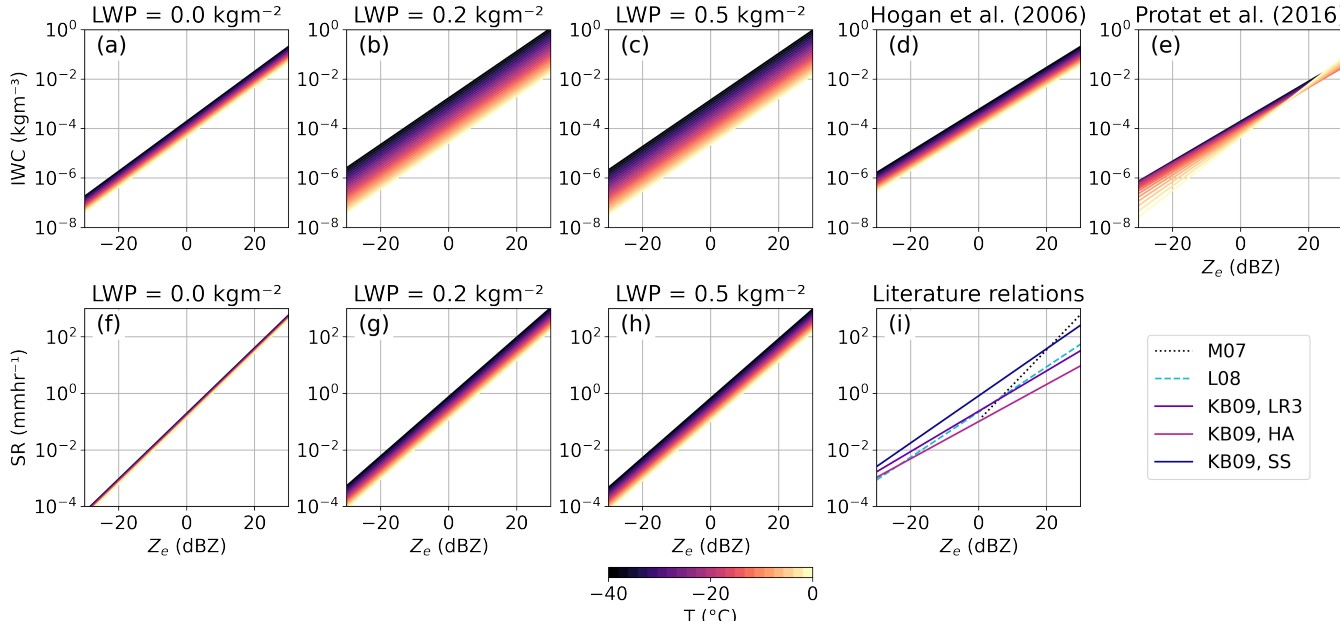

**Figure 5.** IWC-$Z_e$ (top) and SR-$Z_e$ (bottom) empirical functions for (a)&(f) LWP = 0.0 kgm$^{-2}$, (b)&(g) 0.2 kgm$^{-2}$, (c)&(h) 0.5 kgm$^{-2}$ and $T$ ranging from -40° C to 0° C. Empirical IWC-$Z_e$ functions from Hogan et al. (2006) and Protat et al. (2016) are shown in (d) and (e), respectively. SR-$Z_e$ from Matrosov (2007) (M07), Liu (2008) (L08), and Kulie and Bennartz (2009) (KB09, LR3; KB09, HA; KB09, SS) are shown in (i).

et al. (2016); Matrosov (2007); Liu (2008); Kulie and Bennartz (2009) are included for comparison. Here, we applied our viewing angle correction to compute $Z_e$ for 40° observations. At constant (W-band) $Z_e$, IWC and SR increase with decreasing temperature. This is similar to Hogan et al. (2006) and Protat et al. (2016) for $Z < 20\,\text{dBZ}$. Because we lack data points with $Z > 20\,\text{dBZ}$, we cannot confirm the inverse temperature behavior for such large $Z_e$ values. Our SR-$Z_e$ relations follow Matrosov (2007) more closely than to the others shown here.

Different threshold values for LWP to set $q_4$ and $q_8$ to zero—thereby excluding LWP from the IWC and SR relations—were tested and 0.1 kgm$^{-2}$ offered the best trade-off between improvement in Pearson correlation (R$^2$) and RMSE, while assuring a sufficient amount of data with LWP above the threshold (about 26% of SAIL data).

## 4.2 Validation and uncertainty estimates

The empirical relations presented in Sect. 3.4 are validated based on data from SAIL data and additional mid- and high-latitude ground-based sites in Hyytiälä, Ny-Ålesund, and Eriswil (Sect. 2) and compared to literature IWC-$Z_e$ relations from Hogan et al. (2006) and Protat et al. (2016). We first demonstrate the application to ground-based radar using the original vertical resolution of the respective instrument (Sect. 4.2.1). Second, we investigate the application to space-borne radar using the example of the planned WIVERN mission (Sect. 4.2.2). We compare IWC and SR derived with our empirical relations using





reflectivity $Z_e$ from the lowest range bin to the reference IWC and SR derived from in situ data (Sect. 3.2). IWC and SR derived with Eq. 6-9 relations are denoted $\text{IWC}_{\text{regression}}$ and $\text{SR}_{\text{regression}}$, respectively. Reference IWC and SR based on in situ data (Eq. 4 and Eq. 5) are denoted $\text{IWC}_{\text{reference}}$ and $\text{SR}_{\text{reference}}$, respectively. We further derive the normalized root mean

square error (NRMSE) as a function of IWC and SR, respectively, and compare SR results to gauge measurements at SAIL and Hyytiälä (Sect. 4.2.3).

### 4.2.1   Application to ground-based radar

Figure 6 shows a 2d density plot of $\text{IWC}_{\text{regression}}$ vs. $\text{IWC}_{\text{reference}}$ for SAIL data and all additional sites for the empirical functions using $M$ as well as using LWP. Including $M$ gives a high Pearson correlation coefficient for SAIL data and all

sites of about $R^2$=0.96. Without knowledge of $M$—i.e., when there are no in situ measurements at a given cite—LWP can act as a proxy of riming, reducing uncertainties compared to using only $Z_e$ and $T$. Fig. 6.c shows that the relation from Hogan et al. (2006) overestimates IWC for our data. This is likely due to Hogan et al. (2006) using mass-size parameters for unrimed particles in there calculations of reference IWC. As shown in Fig. 4, unrimed particles have higher IWC at the same $Z_e$ as rimed particles. Therefore, applying an IWC relation derived for unrimed particles to data including riming leads to

an overestimation of IWC. Protat et al. (2016) performs better (Fig. 6.d), but also shows a slight overestimation compared to our relations, especially for small IWC. Our relations have higher $R^2$, lower RMSE, and ME closer to zero than the literature relations. $R^2$, RMSE, and ME were derived over the whole IWC range to compare the different relations rather than give uncertainty estimates of IWC, as discussed later in Sect. 4.2.3.

Similarly, Fig. 7 shows the performance of our SR-$Z_e$ relations compared to the reference SR. Again, the relation including

$M$ outperforms the one with LWP, but the difference is less drastic. This is likely due to high fall velocities at large $M$ counteracting the effect of riming on IWC at constant $Z_e$. While the IWC relations developed for SAIL perform similarly well for the other sites, the SR relations performs noticeable worse indicating site-specific effects. However, the largest density of data falls along the 1:1 line and slightly negative ME close to 0 indicate only a small negative bias.

### 4.2.2   Application to space-borne radar

We use the measurement geometry of the planned WIVERN (Illingworth et al., 2018; Battaglia et al., 2022) instrument as an example to demonstrate the application of our empirical relations to space-borne radar. WIVERN will be equipped with 94 GHz radar and a passive 94 GHz radiometer observing profiles of $Z_e$ and brightness temperature $T_B$ at an incidence angle of close to 40°. A LWP retrieval using the $T_B$ data in a similar approach to Ruiz-Donoso et al. (2020) and Billault-Roux and Berne (2021) is planned. To approximate WIVERN $Z_e$ observations, the high resolution, ground-based data from SAIL and the

additional sites are down-sampled to WIVERN geometry, i.e., a vertical resolution of about 580 m and a horizontal resolution of 1 km. In addition, uncertainty estimates are applied to approximate WIVERN measurements. $Z_e$ uncertainties are derived based on simulations (Battaglia et al., 2024); for $T$ an uncertainty of 2 K, and for LWP an uncertainty of $30\,\text{g m}^{-2}$ are assumed. $30\,\text{g m}^{-2}$ was chosen based on the maximum uncertainty of the retrievals from Ruiz-Donoso et al. (2020) and Billault-Roux and Berne (2021) (in mid- and high-latitudes). We also test assuming a higher LWP uncertainty of $60\,\text{g m}^{-2}$. We don't show



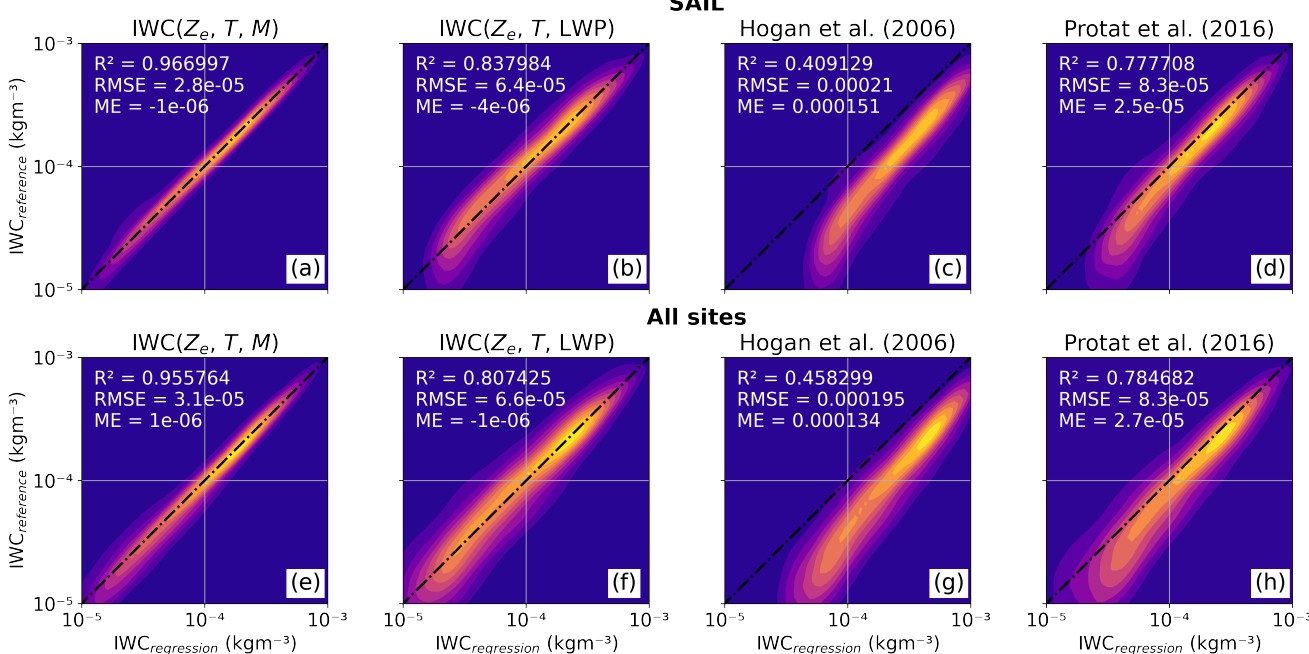

**Figure 6.** 2d density plot of IWC in $\mathrm{kgm^{-3}}$ derived with empirical functions from (a)&(e) equivalent radar reflectivity $Z_e$, air temperature $T$, and normalized rime mass $M$, and (b)&(f) from $Z_e$, $T$, and liquid water path LWP ($\mathrm{IWC_{regression}}$) vs. in situ measurements ($\mathrm{IWC_{reference}}$), which have been used to derive the empirical functions, for the SAIL site (top), and all sites (bottom). (c)&(g) and (d)&(h) show the performance of literature relations from Hogan et al. (2006) and Protat et al. (2016), respectively, where the $Z_e$ was corrected for the viewing angle. Pearson correlation coefficient $R^2$, root mean square error (RMSE), and mean error (ME) derived for the linear IWC data are displayed in the left corner of each subpanel. The 1:1 line is shown as a black, dash-dotted line. Data point density is plotted in ten levels from lowest (blue) to highest (yellow).

the performance of Eq. 6 and Eq. 7, because currently, methods to derive $M$ without in situ data do not exist. Methods based on Doppler velocity (Mosimann, 1995; Kneifel and Moisseev, 2020; Mason et al., 2018) are not applicable in complex terrain due to orography-induced vertical air motions. The method by Vogl et al. (2022) would need to be calibrated for $M$ first.

For space-borne application, the spread in IWC-$Z_e$ space is larger than for ground-based data as is expected. However, the Pearson correlation coefficient is still reasonably high with $R^2$=0.66, even when we apply our empirical relations to all sites.
Doubling the LWP error from $30\,\mathrm{gm^{-2}}$ to $60\,\mathrm{gm^{-2}}$ has barely any impact, because other error sources dominate the resulting variability. The relations from Hogan et al. (2006) and Protat et al. (2016) again result in an overestimation of IWC with the latter performing better applied to our data.

Unsurprisingly, a larger spread for space-borne than ground-based is also present for our SR-$Z_e$ relations. While the application to SAIL data results in higher $R^2$, lower RMSE, and lower ME than to the other sites, the largest density of data is close




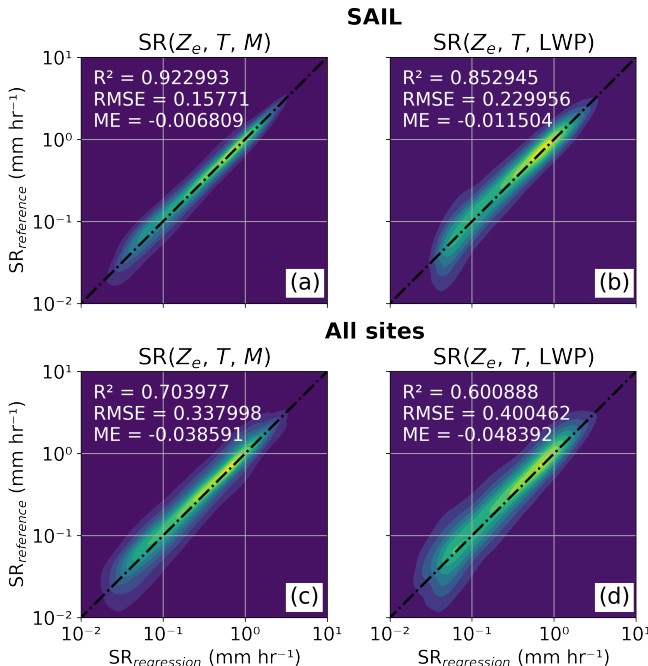

**Figure 7.** Same as Fig. 6 (a-b)&(e-f) but for liquid water equivalent SR.

to the 1:1 line and the increased spread predominately occurs for small SR. Positive ME show a slight positive bias, however the bias is small.

### 4.2.3 Error estimation and comparison to gauge measurements

In the previous section, we used RMSE derived over the whole IWC and SR ranges, respectively, to compare the performance of different relations. However, RMSE typically increases with increasing IWC and SR, thus deriving it over the whole IWC and SR ranges does not quantify their respective uncertainties well. The normalized RMSE as a function of IWC and SR, respectively, is a better tool to quantify uncertainties of our relations (Fig. 10). Here, we calculate RMSE of IWC (SR) for 20 logarithmic bins between $0.01\,\mathrm{gm}^{-3}$ and $1\,\mathrm{gm}^{-3}$ ($0.1\,\mathrm{mmhr}^{-1}$ and $10\,\mathrm{mmhr}^{-1}$) excluding bins with less than 150 data points. We define NRMSE as RMSE divided by the center of each bin. As expected, NRMSE are generally lower when applying our relations to ground-based data than to space-borne data and decrease with increasing IWC and SR, respectively. Using $M$ in the IWC function, NRMSE is below 75% over the whole IWC range and below 25% for IWC$>0.1\,\mathrm{gm}^{-3}$ outperforming the 40%-70% NRMSE range reported in Protat et al. (2016) for IWC$>0.05\,\mathrm{gm}^{-3}$. Using LWP, low IWC values close to $0.01\,\mathrm{gm}^{-3}$ have NRMSE of over 150%. NRMSE decreases with increasing IWC getting below 50% for IWC$>0.2\,\mathrm{gm}^{-3}$. For SR, NRMSE for both $M$ and LWP dependent relations are in a similar range. For SR$>0.2\,\mathrm{mmhr}^{-1}$, the $M$ relation results in NRMSE below 60% and the LWP relation below 70% and 80% for ground-based and space-borne application, respectively. While SR NRMSE generally decreases for both relations, there is more variability than for IWC. This is likely due to multiple reasons. First, high





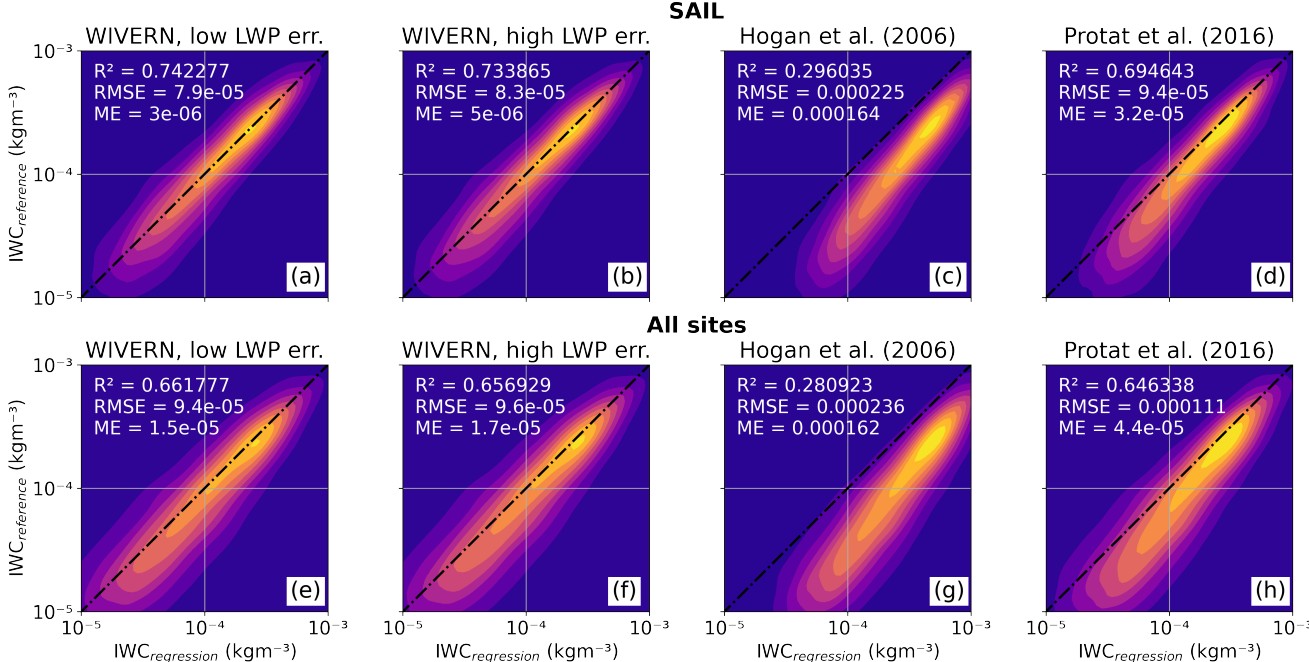

**Figure 8.** As in Fig. 6 but for approximated WIVERN observations (i.e., WIVERN geometry and uncertainty estimations). (a)&(e) and (b)&(f) show results for assuming a LWP uncertainty of $30\,\mathrm{g\,m^{-2}}$ (low) and $60\,\mathrm{g\,m^{-2}}$ (high), respectively.

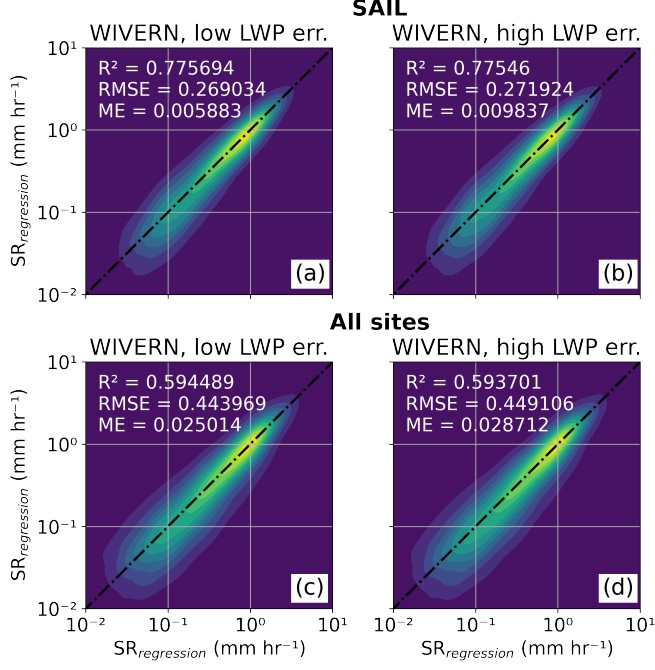

**Figure 9.** As in Fig. 8 (a-b)&(e-f) and but for liquid water equivalent SR.



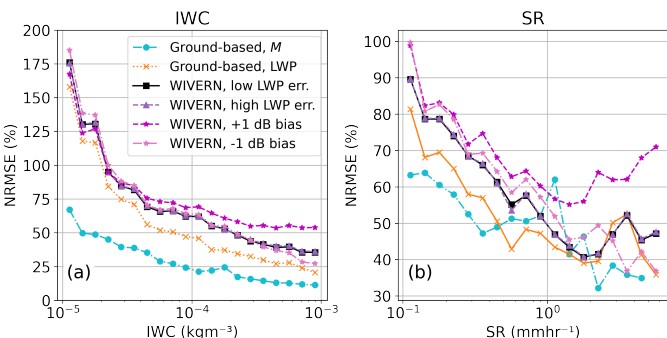

**Figure 10.** Normalized root mean square error (NRMSE) in percentage as function of ground-based and space-borne (WIVERN) estimates of (a) IWC and (b) SR, respectively. NRMSE for positive and negative bias in $Z_e$ of 1 dB are shown in magenta and pink, respectively. Note the different y-axis scales.

SR events are rare and therefore the number of data points for the highest SR bins is lower than for the highest IWC bins. Second, the variability of fall velocities of particles during events with large SR might be larger. The resulting uncertainty of particle fall velocities is likely not covered by our relations. Fig. 4 also shows that at low $Z_e$, meaning generally lower SR, there is a clear spread dependent on particle riming with larger values of $M$ resulting in lower SR. However, at large $Z_e$ (about

>5 dBZ) and therefore generally larger SR, this spread is less visible with lower SR occurring also when particles are close to unrimed. In addition, we tested the performance for a bias in $Z_e$. If $Z_e$ would be biased by $+1$ dB e.g., due to an imperfect calibration, NRMSE were increased by 13 and 16 percentage points on average for IWC$>0.1$ gm$^{-3}$ and SR$>1.0$ mmhr$^{-1}$, respectively. A bias of $-1$ dB is negligible.

SR$_{\text{regression}}$ is also validated against gauge measurements SR$_{\text{gauge}}$, which act as a completely independent reference. The

validation is performed for SAIL data (Fig. 11) and a subset of HYY data (Dec 2023 to Feb 2024, Fig. 12), due to limited data availability. Gauge snowfall measurements can be subject to various sources of errors and gauge derived SR can vary significantly between identical instruments (e.g., Yang and Simonenko, 2014) even though the one in HYY is a operated as a Double Fence Intercomparison Reference (DFIR, Rasmussen et al., 2012) and the one at SAIL was located in a Low Porosity Double Fence (LPDF, Kochendorfer et al., 2023). A 1:1 fit is therefore not expected. However, hourly accumulated SR$_{\text{regression}}$

show no systematic biases compared to SR$_{\text{gauge}}$.

## 5 Conclusions

In this study, we present novel ice water content - equivalent radar reflectivity (IWC-$Z_e$) and snowfall rate - equivalent radar reflectivity (SR-$Z_e$) relations for 40° slanted and vertically pointing W-band radar. We investigate the dependence of IWC-$Z_e$ and SR-$Z_e$ on riming, which we quantify with the normalized rime mass $M$ (Seifert et al., 2019; Maherndl et al., 2024a), and

use $M$ in our relations to reduce the spread in the IWC-$Z_e$ and SR-$Z_e$ spaces. In addition, we present relations using liquid water path (LWP) instead of $M$, which can act as a proxy for the occurrence of riming. LWP is typically easier to measure





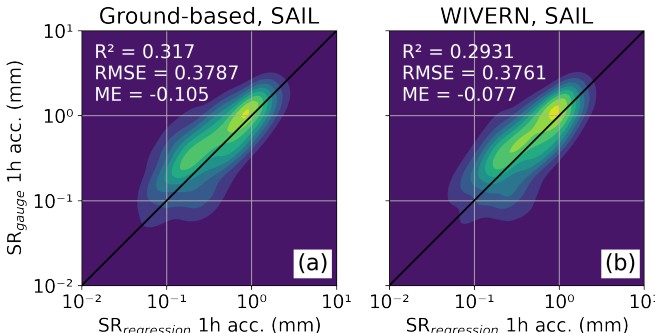

**Figure 11.** 2d density plot of hourly accumulated SR in mm (liquid water equivalent) derived with empirical function from equivalent radar reflectivity $Z_e$, air temperature $T$, and liquid water path LWP ($SR_{regression}$) applied to (a) ground-based radar and (b) the approximated WIVERN measurements vs. gauge measurements ($SR_{gauge}$) for the SAIL site. Data point density is plotted in ten levels from lowest (blue) to highest (yellow).

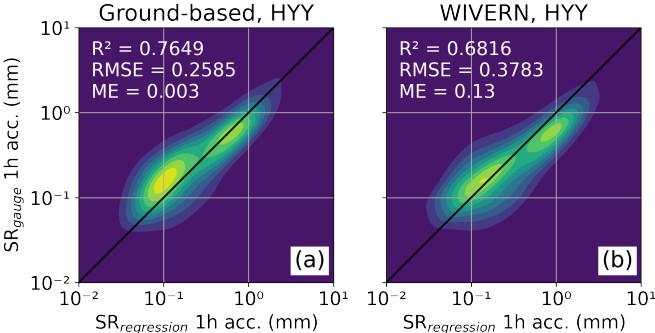

**Figure 12.** As in Fig. 11 but for Hyytiälä, Finland for the time period of Dec 2023 to Feb 2024.

than $M$ so that the relations with LWP can be applied to ground-based or space-borne radar-radiometer instruments. The applicability of the method to observations of the proposed Earth Explorer 11 candidate mission WIVERN (Illingworth et al., 2018; Battaglia et al., 2022) is investigated.

We used joint in situ snowfall (VISSS) and remote sensing (W-band radar and radiometer) data from ground-based sites in mid- and high-latitudes to build a dataset of reference IWC and SR. Reference IWC and SR from the SAIL site (Colorado, USA) are used to derive the IWC-$Z_e$ and SR-$Z_e$ relations, while reference IWC and SR from additional sites in Hyytiälä (Finland), Ny-Ålesund (Svalbard), and Eriswil (Switzerland) are used for validation. In addition, gauge measurements from SAIL and Hyytiälä are used as an independent reference for validation.

Our main findings are summarized in the following:

1. We found that slanted W-band $Z_e$ observations at 40° are $2.29 \pm 0.39\,\text{dB}$ (mean $\pm$ standard deviation) lower than vertically pointing $Z_e$ for constant IWC (Fig. 3). This offset is nearly constant over the full IWC range and likely due to snow



particles being aligned predominately horizontally. As a result, this offset can be applied to correct $40°$ $Z_e$ to $90°$ $Z_e$ and vice versa.

2. For a constant $Z_e$, the IWC is generally lower as the ice and snow particles are more heavily rimed (Fig. 4). This also holds for SR. However, at larger $Z_e$ (about >5 dBZ), the dependence on riming is less pronounced and lower SR also occur for unrimed particles at constant $Z_e$. This is likely due to rimed particles typically having larger fall speeds, thus increased SR, and to more variability in particle fall speed during high SR events in general.

3. We demonstrated the application of our IWC-$Z_e$ and SR-$Z_e$ relations to ground-based sites (Fig. 6 and Fig. 7). When
estimates of $M$ are available, IWC and SR can be derived accurately with Eq. 6 and Eq. 7 ($R^2$=0.96 and $R^2$=0.70 for IWC and SR, respectively). Normalized root mean square error (NRMSE) are below 50% and 25% for IWC>0.01 $\mathrm{g m^{-3}}$ and IWC>0.1 $\mathrm{g m^{-3}}$, respectively. For SR, the NRMSE is below 70% over the SR range. At sites without in situ data, which is currently needed to derive $M$, LWP can act as a proxy for the occurrence of riming (Eq. 8 and Eq. 9) resulting in $R^2$=0.81 and $R^2$=0.60 for IWC and SR, respectively. NRMSE are below 150% and 75% for IWC>0.01 $\mathrm{g m^{-3}}$ and
IWC>0.1 $\mathrm{g m^{-3}}$, respectively, and below 70% for SR>0.2 $\mathrm{mm hr^{-1}}$.

4. We also showed the application of the LWP-dependent formulas to space-borne instruments using the example of the planned WIVERN mission (Fig. 8 and Fig. 9). We approximated future WIVERN measurement by averaging the ground-based data to the coarser WIVERN resolution and applying error estimates consistent to the expected performance of WIVERN. NRMSE of the IWC and SR estimates are less than 10 percentage points higher than for ground-based
applications even when assuming a high estimate for the LWP error (Fig. 10).

5. Comparing our SR estimates to gauge data for SAIL and Hyytiälä shows no stark bias towards over- or underestimation (Fig. 11 and Fig. 12). This strengthens the validity of our relations for different sites.

It must be noted that there are several assumptions that go into deriving the reference IWC and SR data. IWC and SR are based on VISSS observations and assumptions about the mass-size relation of snow particles. The assumed mass-size
parameters were selected for $M$ derived for each time step assuming a mixture of particle shapes. The $M$ retrieval assumes that VISSS observations at the ground are representative of the matched radar volume close to ground. The retrieval method uses forward simulations with PAMTRA and scattering and physical properties of rimed ice particles are based on simulated rimed aggregates. It is assumed that the simulated rimed aggregates are representative of snow and ice particles in nature. Further observational studies focusing on particle mass and scattering behavior are needed to investigate these assumptions.
Our empirical functions were derived and validated based on few sites in mid- and high latitudes in the Northern hemisphere. More sites with combined in situ and W-band radar measurements would be necessary to investigate if the empirical relations can be applied globally.

In conclusion, the proposed IWC and SR relations provide a novel way to reduce uncertainties of IWC and SR estimates for W-band radar by accounting for particle riming. Advantages to current literature relations are the flexibility in terms of viewing



angle (40° slanted and 90° vertical) and the inclusion of LWP, allowing the application to ground-based and space-borne radar-radiometer combinations like EarthCARE or the proposed WIVERN mission. The Doppler capabilities of EarthCARE might even allow to quantify riming from the hydrometeor fall velocities via the approach from Mosimann (1995) or via optimal estimation techniques (Mroz et al., 2023; Mason et al., 2023). Then, the IWC and SR relations including $M$ which have lower uncertainties than the ones based on LWP.

*Data availability.*  SAIL data were obtained from the Atmospheric Radiation Measurement (ARM) user facility, a U.S. Department of Energy (DOE) Office of Science user facility managed by the Biological and Environmental Research Program.: LIMRAD94 (https://doi.org/10.5439/2229846, last access: 28 Nov 2024), VISSS (https://doi.org/10.5439/2278627, last access: 5 Dec 2024), the meteorological in situ data of AMF2 (https://doi.org/10.5439/1786358, last access: 28 Nov 2024), and the microwave radiometer retrieval products (https://doi.org/10.5439/1027369, last access: 5 Dec 2024). Cloudnet data from Hyytiälä, Ny-Alesund, and the PolarCAP campaign are available for
download from https://cloudnet.fmi.fi (last access: 28 Nov 2024). VISSS1 and VISSS2 data from Hyytiälä and Ny-Alesund are published on PANGAEA (HYY: https://doi.org/10.1594/PANGAEA.959046, last access: 5 Dec 2024; NYA: https://doi.org/10.1594/PANGAEA.958537 and https://doi.org/10.1594/PANGAEA.965766, last access: 5 Dec 2024). VISSS3 data from Hyytiälä and VISSS1 data from PolarCAP are available upon request.

## Appendix A:  Riming dependent SSRGA coefficients for 40° slanted radar

We performed the same analysis as in Maherndl et al. (2023) to parameterize the Self-Similar Rayleigh-Gans Approximation (SSRGA, Hogan and Westbrook, 2014; Hogan et al., 2017) parameters $\alpha_e$, $\kappa$, $\gamma$, $\beta$, and $\zeta_1$ but for 50° tilted instead of horizontally aligned particles to account the 40° observations during SAIL in our scattering calculations. For further detail in regards to SSRGA and the riming-dependent parametrization, we refer to Maherndl et al. (2023).

Eq. A1 gives the form of the function to derive each SSRGA parameter.

$$\text{SSRGA parameter} = p_1\, M^{2p_0} + p_2\, M^{p_0} + p_3, \tag{A1}$$

where $p_i$ are fit coefficients.

We obtain the following parameterizations of the SSRGA parameter depending on $M$ for 40° slanted radar:

$$\alpha_\mathrm{e} = 0.0168\, M^{1.007} + 0.1609\, M^{0.5035} + 0.7234, \tag{A2}$$

$$\kappa = 0.117\, M^{1.007} - 0.0022\, M^{0.5035} + 0.0429, \tag{A3}$$

$$\gamma = -0.8126\, M^{1.007} + 1.6618\, M^{0.5035} + 2.4369, \tag{A4}$$



$$\beta = -2.648\,M^{1.007} + 0.6949\,M^{0.5035} + 2.8542, \tag{A5}$$


$$\zeta_1 = 0.1125\,M^{1.007} - 0.1316\,M^{0.5035} + 0.1158. \tag{A6}$$

## Appendix B: Slanted vs. vertical $Z$ during SAIL

Fig. B1 shows distributions of $Z_e$ close to ground during scans in February for 40° slanted and vertical observations. The correction derived in Sect. 3.3 shifts the 90° distribution closer to the 40° distribution, especially for the higher reflectivity right
edge. Median and quantile values of the 40° and the corrected 90° data show close agreement strengthening the validity of our correction.

*Author contributions.* NM developed the described methods, analyzed and plotted the data, and wrote the manuscript. AK collected and processed the LIMRAD94 data from SAIL and PolarCAP. MM and AB acquired funding and guided the research project. All authors reviewed and edited the manuscript.

*Competing interests.* At least one of the (co-)authors is a member of the editorial board of Atmospheric Measurement Techniques.

*Acknowledgements.* This work was supported by the European Space Agency under the activity WInd VElocity Radar Nephoscope (WIVERN) Phase A Science and Requirements Consolidation Study (ESA Contract Number 4000144120/24/NL/IB/ab) and by three projects funded by the Deutsche Forschungsgemeinschaft (DFG, German Research Foundation): 408008112 ("Characterization of orography-influenced riming and secondary ice production and their effects on precipitation rates using radar polarimetry and Doppler spectra" (CORSIPP) within
the Priority Program SPP 2115 "Polarimetric Radar Observations meet Atmospheric Modelling (PROM) – Fusion of Radar Polarimetry and Numerical Atmospheric Modelling Towards an Improved Understanding of Cloud and Precipitation Processes"), 268020496 (TRR 172 "Arctic Amplification: Climate Relevant Atmospheric and Surface Processes, and Feedback Mechanisms" (AC)[3], and 516261703 ("Evaluating Microphysical Pathways Of midlatitude Snow formation (EMPOS)"). We acknowledge ACTRIS and Finnish Meteorological Institute for providing the data set which is available for download from https://cloudnet.fmi.fi. This research was supported by the Atmospheric
Radiation Measurement (ARM) user facility, a U.S. Department of Energy (DOE) Office of Science user facility managed by the Biological and Environmental Research Program. We thank Mario Montopoli from CNR Rome for his constructive feedback, which helped to improve this study.



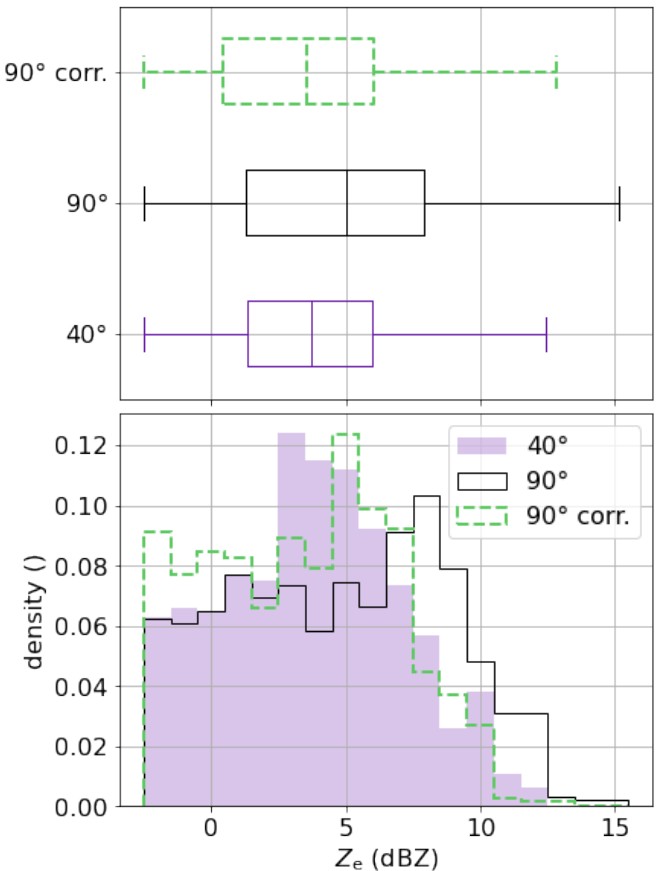

**Figure B1.** Boxplots (top) and distributions (bottom) of W-band $Z_e$ during scans in Feb 2023 at SAIL at 40° (purple), 90° (black), and 90° corrected to 40° using the correction from Sect. 3.3 (green, dashed).

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
