# Peer review of "Riming-dependent Snowfall Rate and Ice Water Content Retrievals for W-band cloud radar"

_EGUsphere, 2024_

## Referee Comment (RC2)

The paper presents the development of emperical IWC-Ze and SR-Ze relationships for obtaining snowfall rate and ice water content from W-band radar observations and accounting for the effects of riming. The work documents the performance of the relationships both when the normalized rime mass is known and when liquid water path is used as a proxy for rime mass. The normalized root-mean-square errors are somewhat large for small IWCs and SRs, but decrease as IWCs and SRs increase, giving performance generally better than a number of previously published studies.

The presentation of the work is clear and well organized. The conclusions are supported by the results. I have some requests for clarification or additional explanation that I consider mostly minor; see the comments below. I'm recommending acceptance with minor revisions.

My most significant concern is that it took a fair amount of examination of prior work (Seifert et al., 2019 and the earlier works by Maherndl et al.) to resolve in my mind the implications of changes in normalized rime mass. For example, does an increase in normalized rime mass mean that mass and D_max are unchanged,but that some of the mass is converted to lower-density rime mass? It would be helpful for the authors to provide a brief explanation to provide background for the readers, then allow them to consult the references if they desire a deeper understanding.

Science and technical comments
===============================

L 21-22: Most gauges do not measure snowfall rate directly. Normally, they measure accumulation over a time interval, which is then used to compute mean liquid-equivalent snowfall rate over the time interval.

L 28-29: I don't think that W-band radars are commonly used for snowfall. There are complications introduced by attenuation and non-Rayleigh scattering by snow particles. For many operational applications, C- or S-band weather radars are the most commonly used for estimating snowfall rates, usually using empirical Ze-S relationships. Space-based operational missions like Global Precipitation Measurement and Tropical Rainfall Measurement Mission use Ku- and Ka-band radars, although they do suffer from a lack of the sensitivity needed for lighter precipitation. Research programs like CloudSat and US Department of Energy's Atmospheric Radiation Measurement do employ W-band and Ka-band radars which are used for snowfall retrieval, but W-band radars are probably the least common types used for snowfall.

L 34-35: Do you have a reference for the information content of satellite observaions for snowfall?

L 56-58: After reviewing Scarsi et al. (2024), I don't see any claims or documentation that the 800 km swath of WIVERN produces reduced uncertainty in polar snowfall estimates versus CloudSat. Please be more clear about what your are claiming here.

L 59-60: I'm concerned about this assertion since it seems at odds with what has been found for the scanning Ku- and Ka-band radars used by GPM (scattering from the beam side lobes causing a deeper blind zone at larger angles of incidence). See Kubota et al. (2016, JTECH, doi:10.1175/JTECH-D-15-0202.1). The Coppola et al. reference is not available for examination.

L 97-98: Do you have a reference for the statistical retrieval itself? Is this the LWP from the two-channel or from the three-channel radiometer? How was the Pluvio gauge fenced?

L 160: It seems rather arbitrary to remove light snowfall cases. What is the justification for doing this?

L 173-174:  When you say "horizontally aligned", do you mean perfectly horizontally aligned, or are the particle orientations allowed to vary randomly by some maximum angle relative to horizontal (i.e., 'flutter')?  My experience with discrete dipole calculations indicates the first approach can overestimate backscatter cross-sections if the radar is vertically pointing.  This may not be a significant concern with a 40-degree viewing angle, though.

L 180:  Can you provide a reference for this assertion about the dominance of riming for particle mass?

L 219-224:  So, to be sure that I understand, you compared the relationship between Ze and IWC for 90-degree observations from SAIL against the corresponding relationship derived for 40-degree observations from SAIL?  And the m(D) parameters by which IWCs were determined were obtained by fitting modeled reflectivities (obtained from the observed PSDs) against the observed 40-degree slant path observed Ze?

L 232-249:  There are a few things to clarify here.  First, although the relationships given in (6), (7), (8), and (9) are general, is it your intention to fit these relationships using the 40-degree Ze observations?  And I'm fairly certain the 'LWP' used here is the LWP measured in the vertical direction, but it would be helpful to explicitly say so.  The reason to clarify this is that at least one of the DOE microwave radiometers (the two-channel) produces both a liquid water path in the zenith direction and a liquid water path that is along the line-of-sight path of the radiometer.

L 273-275, Figure 4:  Distinct point shapes are not apparent, particularly in the locations where the scatter plot point density is high.  Would 2D histograms be more useful for panels (a) and (c)?

Also, in Figure 4, there are some interesting behaviors.  In panel (a), if one follows a vertical, downward trajectory of constant Ze, normalized rime mass M increases while IWC decreases.  It would be interesting to see what this means in terms of size distribution changes, but that is beyond the scope of this work.

L 297-299:  This explanation seems a bit simplistic.  Could there not be differences versus particle habit assumptions or versus the population of PSDs used by Hogan et al. to produce their relationship?  It's not apparent to me that this difference is attributable solely to the use of unrimed particles by Hogan et al.

L 314-317:  At the SAIL site, it appears the LIMRAD94 was limited to a range of 2000m, or a height above ground of about 1300 m at the 40-degree observation angle.  Similarly the maximum range was 2000m at Eriswil.  The maximum heights for the RPG-FMCW-94-DP and JOYRAD-94 are not specified.

For this WIVERN-like comparisons, did you simulate an atmospheric column?  What were the depths of the columns and how did you treat 94-GHz attenuation?  Also, how did you approximate the 1 km horizontal resolution using the ground-based observations?  Please explain more about how the WIVERN observations are simulated from the ground observations.

Finally, given the empirical nature of the retrieval, it is not clear how these uncertainties are introduced into the retrieval process and how their effects are evaluated.  Please explain.

L 323-326:  See my previous comment regarding how the effects of uncertainties are evaluated.

L 356-360:  Wind effects ('pumping') are probably the most significant source

of error for heated Pluvio guages. Did you consider filtering out cases with high wind conditions to see if the validation results improved?

L 380-383: See my earlier comment. At face value, increased riming of particles should increase IWC. But in this case, you are describing the changes given a constant Ze. So, what physical processes are happening if Ze is constant but riming increases and IWC decreases? It would be helpful to most readers to provide some insight here.

L 391-395: Se my earlier comment. More details are needed in the discussion section regarding how these WIVERN-like measurements were produced and how observational errors were propagated into the empirical retrieval.

Minor language corrections
==========================

L 40: Should be 'particle size distributions'.

L 58: Should be 'signficantly reducing the uncertainty'.

L 295: Should be 'site'.

L 298: Should be 'their'.

L 357: Should be 'HYY is operated as'.

---

## Author Comment (AC1)

**Riming-dependent Snowfall Rate and Ice Water Content Retrievals for W-band cloud radar**

N. Maherndl, A. Battaglia, A. Kötsche, and M. Maahn

March 24, 2025

*Original Referee comments are in italic*

manuscript text is indented, with added text underlined and

We would like to thank the reviewers for their helpful comments. We revised the manuscript and responded to all of the reviewers' comments.

**Reviewer I**

**Short summary**

*The authors derive relations between radar reflectivity (Ze) and ice water content (IWC), as well as Ze and snowfall rate (SR). Compared to existing relations, they take into account the normalized rime mass (M) into their relations. The relations are trained on data from one field site and evaluated on data from three other field campaigns. Additionally, since M is often unknown, they also provide relations based on liquid water path (LWP). Those relations could be applied to space-borne measurements.*

**General Comment**

*I think the manuscript is generally well written and addresses a relevant scientific question within the scope of the journal. Therefore, it is likely worth of publication after all comments are addressed.*

*I like that the authors train their relations based on data from one site and then test the performance at three other locations on very different latitudes. This greatly increases the credibility into the universality and robustness of the method. The only issue I see is the following: The relations are established as a fit of a polynomial function between Ze and IWC. However, IWC can not be observed directly. Since the in situ imager can only measure the particle size distribution N(D), we need an estimate of the snowflake mass. This mass information depends on estimates of the normalized rime mass M, which in turn is derived from N(D) and Ze. Therefore, in the training, the "labels" (IWC) are not independent from the "features" (Ze)!*

*This is the case for the train and test sites. The only truly independent evaluation, if I am correct, is coming from the SR gauge.*

*I think one has to discuss carefully what this double use of Ze implies. Would a bias in Ze go unnoticed, since it affects both sides of the fit equation equally? Can it explain the weaker correlation with gauge measurements? Generally, for me, it seems like the retrievals of IWC and M face a similar problem: In both cases, Ze, which depends on number and mass, has not enough information to fully constrain the values of interest. For IWC, we need additional information about the mass or density (like M).*

*For the retrieval of M, we need information about the particle number, which is coming from the particle imager N(D). Then, a forward operator in combination with an optimal estimation approach is used to find the M, which matches the observed Ze best.*

*So why not using the same approach directly for the retrieval of IWC? I.e., given N(D),*

*using a forward operator and vary IWC until it matches the observed Ze? Otherwise, an interesting study on a unique dataset!*

We thank the reviewer for the positive review and the constructive comments, which helped to improve the manuscript.

Regarding the comment about IWC not being independent from $Z_e$, we have added the following in the revised manuscript:

> It must be noted that our reference IWC and SR data are not fully independent of $Z_e$ because we derive the particle mass from the retrieved normalized rime mass $M$. This is a necessary limitation because IWC and SR cannot be inferred from the available in situ measurements alone. For SR, we evaluate our approach with completely independent SR gauge measurements.

To give a more detailed answer here, yes, our reference data of IWC and SR are not independent from $Z_e$. We are aware that this is a limitation of the approach, however this way, we get the best estimate of IWC given the available data. An independent IWC estimate is not possible because we currently don't have size resolved measurements of particle mass. The radar data we use is quality controlled and we assume $Z_e$ is not biased. If $Z_e$ would have a positive bias, then $M$ would have a positive bias as well, resulting in a positive bias of IWC.

The suggestion to directly retrieve IWC would result in the excact same outcome as the $M$ retrieval. This is because in the forward operator, particle mass is a function of $M$ (because we use the mass-size relation form Maherndl et al. (2023)).

**Other comments**

*line 36-42: I believe error cancellation happens if the relations are trained on enough data to capture the full snowfall climatology with all its diversity in shape, habit, etc. Then, a relation can be wrong in a single case, but the overall average might be correct. However, if the relations themselves differ by about one order of magnitude, why would you assume that the error cancels out on seasonal time scales? If one relation is systematically higher than another (compare e.g. in Fig. 5 i) ), it will also lead to systematically higher results for any reflectivity time series!*

We agree that error cancellation results when relations are trained on a large amount of data capturing the snowfall climatology. We added:

>  Relations can have significant uncertainties for individual cases,  but are successfully applied to space-borne radar data sets because the random errors cancel  out partly in seasonal time

scales assuming they are trained on a large enough data set to capture the full snowfall climatology (Kulie, Bennartz, 2009).

*Line 205: Why not discarding NaN cases? (Would it reduce the number of data points too much?)*

Yes, the number of data points would be heaviliy reduced. There are often size bins with a number concentration but no fall velocity in the VISSS products due to the different methods in deriving number and velocity distributions. We added:

> ... To avoid unrealistic behavior at the edges of the size spectrum, NaN values of $v$ are filled with $v$ from the closest available size bin. Removing cases with NaN values would greatly reduce the number of data points.

*Sec. 3.4, Equations 6,7,8,9: Can you motivate, why you chose exactly this functional form as a basis for the regression? For example, when I look at the measurement data in Fig. 4a+c, it seems like the cone of data points is getting more narrow towards the right (the spread of IWC and SR is becoming less with higher Ze). This is not visible in your fit functions in Fig. 4b+d. Why not including a coefficient in the fit function which would allow for this flexibility?*

We opted for a log-log IWC-Z and SR-Z functional form due to its common practice. Temperature in degree Celsius must be included linearly due to negative values. $M$ is included logarithmically, because the distribution of $M$ often follows a Gaussian shape in logarithmic space. The smaller spread of IWC and SR at higher $Z_e$ could be due to the rarer occurrence of such high $Z_e$ in our data. It could be the case that if we were to have a larger data set, the spread at high $Z_e$ were also larger. In the revised text, we added:

> We chose the functional forms of Eq. 6-9, because power law relations are commonly used for IWC-$Z_e$ and SR-$Z_e$ (Fuller et al., 2023). $T$ in degree Celsius must be input linearly due to negative values. The logarithm of $M$ and LWP are used, because the logarithm of both variables often follows a Gaussian shape.

*Line 224: Why not using the SAIL-derived 2.25 dB as offset for the viewing angle correction, instead of the 2.29 dB derived from inter-site comparison (line 218)? That way, the data evaluation from the validation sites would be completely independent from the training site.*

Because of the small data set when only using SAIL data and the limitation of which $M$ threshold we can use for unrimed particles (see lines 229-231 of the revised manuscript). Also, the standard deviation for SAIL only is much larger due to the small data amount.

*Fig. 3 caption: I thought the offset between SAIL and HYY+NYA is 2.29 dB +-0.39?*

Yes, this was a mistake. We fixed it.

*Line 276: Does this mean the relations become uncertain/invalid beyond 20 dBZ? I would recommend to plot only the range where data points are available, or at least indicate the range where the relations are based on extrapolation.*

We have added shading for $Z_e$ larger 15 dBZ indicating the range of available data in Fig. 4.

*Line 289: "based on in situ data" − > not only (see main comment)*

We added:

> Reference IWC and SR based on in situ data and retrieved normalized rime mass $M$ (Eq. 4 and Eq. 5) are denoted IWC$_{reference}$ and SR$_{reference}$, respectively.

*Line 307: Site specific effects − > what could those be?*

Orographically induced turbulence at the SAIL site, for example. We added:

> While the IWC relations developed for SAIL perform similarly well for the other sites, the SR relations performs noticeable worse indicating site-specific effects (e.g., orographically induced turbulence might affect snowfall at the SAIL site).

*Sec. 4.2.2.: You trained the retrieval based on air temperature measurements and, if I understand correctly, also use air temperature in your "simulated" satellite retrieval (+-2K uncertainty). For the case of a real satellite, which temperature would you use? Would it be valid to use brightness temperature? Would this limit the retrieval to cloud top?*

For WIVERN, we suggest using auxiliary data from ECMWF as input for air temperature. The retreival can therefore be applied to the whole column, not only cloud top. This is common praxis also for other satellite missions such as CloudSat.

*Line 325: Interesting. I would have assumed, since the inclusion of LWP (as proxy for M) is the main improvement over existing relations, the influence of LWP would be bigger. What are the other error sources you mention?*

Including LWP leads to an improvement as opposed to not including LWP even when assuming a large uncertainty. However, it is no surprise that the literature relations perform worse than our relations, because they were derived from different data sets. Errors result from the natural variability in IWC-$Z_e$ space, the random errors we included to approximate WIVERN observations, and the regridding to the coarser WIVERN resolution. We added:

Doubling the LWP error from $30\,\mathrm{g\,m^{-2}}$ to $60\,\mathrm{g\,m^{-2}}$ has barely any impact, because other error sources (e.g., from averaging to the WIVERN resolution) dominate the resulting variability.

*Fig. 7: Instead of SAIL vs All sites, I would probably split SAIL vs Other sites (clear Train vs Test distinction). Also in other figures.*

We changed the figure row title from *All Sites* to *Other sites* and changed *all sites* to *all other sites* in the text.

*Line 352: A positive bias increases the NRMSE, a negative has no effect. Why is this asymmetry?*

The asymmetry is due to the logarithmic scale.

*Fig. 11, 12: It is interesting to see that the gauge error (R2, RMSE, ME) seems to be generally less for HYY than for SAIL, even though the relations are developed on SAIL data.*

Yes, this is interesting and could be due to lower variability of snowfall at HYY than SAIL, but this is only speculation.

*Conclusion 1 (line 376 following): Since the viewing angle offset was derived for unrimed particles, high and low IWC situations are mainly different in number concentration and particle size, I assume. Since riming is a main point of this study, it is interesting to ask what happens for riming. Intuitively, at least for strong riming, I would expect the particles to become rounder and therefore, the offset between vertical and slanted observations to become smaller?*

Yes, that is expected for very heavily rimed graupel particle, but not necessarily for light riming. We don't have enough data with very strong riming to prove this hypothesis.

**Technical Comments**

*Line 42: "cancel partly out" − > "cancel out partly"*

*Line 166: "Windows corresponds"− > Windows correspond*

*Line 218: "between the vertically pointing" − > remove?*

*Line 221: "when they when" − > when they where*

*Line 298: there − > their*

Thank you, we fixed the mistakes.

*Figure 4, panel b): What is the black dashed line? Add a legend.*

Thanks for noticing, the line should not be part of the plot and was removed. (It was a leftover from testing.)

*Figure 6: Even though histogram units are often w.r.t. density or similar, a colorbar would be nice.*

We have added colorbars to Fig. 6-9 and Fig. 11-12.

*Fig. 11 & 12 could optionally be combined into one figure, like Fig. 9.*

We decided to show Fig. 12 separately, because it shows HYY data only. The combined figures all depict SAIL vs. all other sites.

*Line 414, last sentence: incomplete?*

Yes, we fixed it to:

> Then, the IWC and SR relations including $M$ can be used, which have lower uncertainties than the ones based on LWP.

---

## Author Comment (AC2)

**Riming-dependent Snowfall Rate and Ice Water Content Retrievals for W-band cloud radar**

N. Maherndl, A. Battaglia, A. Kötsche, and M. Maahn

March 24, 2025

*Original Referee comments are in italic*

> manuscript text is indented, with added text underlined and

We would like to thank the reviewers for their helpful comments. We revised the manuscript and responded to all of the reviewers' comments.

**Reviewer II**

*The paper presents the development of emperical IWC-Ze and SR-Ze relationships for obtaining snowfall rate and ice water content from W-band radar observations and accounting for the effects of riming. The work documents the performance of the relationships both when the normalized rime mass is known and when liquid water path is used as a proxy for rime mass. The normalized root-mean-square errors are somewhat large for small IWCs and SRs, but decrease as IWCs and SRs increase, giving performance generally better than a number of previously published studies.*

*The presentation of the work is clear and well organized. The conclusions are supported by the results. I have some requests for clarification or additional explanation that I consider mostly minor; see the comments below. I'm recommending acceptance with minor revisions.*

*My most significant concern is that it took a fair amount of examination of prior work (Seifert et al., 2019 and the earlier works by Maherndl et al.) to resolve in my mind the implications of changes in normalized rime mass. For example, does an increase in normalized rime mass mean that mass and D_max are unchanged,but that some of the mass is converted to lower-density rime mass? It would be helpful for the authors to provide a brief explanation to provide background for the readers, then allow them to consult the references if they desire a deeper understanding.*

We thank the reviewer for the positive review and the constructive comments, which helped to improve the manuscript.

We added an additional explanation of the normalized rime mass $M$ and its link to particle size.

> $M$ is a quantitative measure of how heavily rimed an ice particle is with $M = 0$ meaning completely unrimed and $M \to 1$ meaning spherical graupel. $M$ is not necessarily dependent on particle size. However, assuming a fixed amount of liquid water available for riming, larger particles will have lower $M$ than smaller particles after riming (Maherndl et al., 2023).

**Science and technical comments**

*L 21-22: Most gauges do not measure snowfall rate directly. Normally, they measure accumulation over a time interval, which is then used to compute mean liquid-equivalent snowfall rate over the time interval.*

We agree and changed the sentence to:

> While gauges provide direct measurements of  SR, they are prone to large uncertainties (e.g., Saltikoff et al., 2015).

*L 28-29: I don't think that W-band radars are commonly used for snowfall. There are complications introduced by attenuation and non-Rayleigh scattering by snow particles. For many operational applications, C- or S-band weather radars are the most commonly used for estimating snowfall rates, usually using empirical Ze-S relationships. Space-based operational missions like Global Precipitation Measurement and Tropical Rainfall Measurement Mission use Ku- and Ka-band radars, although they do suffer from a lack of the sensitivity needed for lighter precipitation. Research programs like CloudSat and US Department of Energy's Atmospheric Radiation Measurement do employ W-band and Ka-band radars which are used for snowfall retrieval, but W-band radars are probably the least common types used for snowfall.*

We disagree when it comes to space-borne snowfall estimates. Until recently, Cloud-Sat provided the reference for global snowfall estimates and EarthCare will likely fill that role. GPM does not cover polar regions in addition to the sensitivty issues you mentioned.

*L 34-35: Do you have a reference for the information content of satellite observaions for snowfall?*

Yes, we added a reference:

> Further, the information content of satellite observations is typically not sufficient to constrain the highly variable microphysical properties of snow and ice particles unambiguously (Wood, L'Ecuyer, 2021).

*L 56-58: After reviewing Scarsi et al. (2024), I don't see any claims or documentation that the 800 km swath of WIVERN produces reduced uncertainty in polar snowfall estimates versus CloudSat. Please be more clear about what your are claiming here.*

Apologies, we linked the wrong Scarsi et al. (2024) paper, we have updated the reference. WIVERN has a higher sampling frequency due to its 800 km swath as compared to the "pencil beam approach" from CloudSat. Scarsi et al. (2024) show that a higher sampling frequency does reduce uncertainty.

*L 59-60: I'm concerned about this assertion since it seems at odds with what has been found for the scanning Ku- and Ka-band radars used by GPM (scattering from the beam side lobes causing a deeper blind zone at larger angles of incidence). See Kubota et al. (2016, JTECH, doi:10.1175/JTECH-D-15-0202.1). The Coppola et al. reference is not available for examination.*

We updated the reference, which is now publicly available as preprint in EGUsphere:

In addition, WIVERN's 42° angle of incidence results in a  smaller radar blind zone near the surface (especially over the ocean)  (Coppola et al., 2025).

*L 97-98: Do you have a reference for the statistical retrieval itself? Is this the LWP from the two-channel or from the three-channel radiometer? How was the Pluvio gauge fenced?*

We used the ARM 3-channel microwave radiometer liquid water path and added in the text:

> We use additional data acquired by ARM of near-surface air temperature $T$, SR from a Pluvio weighing precipitation gauge and liquid water path (LWP).  For the latter, we use the ARM 3-channel microwave radiometer LWP, which is derived from a site-specific statistical retrieval from microwave radiometer brightness temperature measurements.

The Pluvio gauge was located in a Low Porosity Double Fence as written in lines 373-374 (of the revised manuscript).

*L 160: It seems rather arbitrary to remove light snowfall cases. What is the justification for doing this?*

Under extremely light snowfall (corresponding to less than 0.1 mm/hr SR), random errors of the observed PSD are problematic. In addition, the matching between the in situ snowfall camera and the lowest radar range gate is prone to large uncertainty. For example, the radar might "miss" light snowfall that is measured by the in situ instrument. Additionally, light snowfall cases contribute less to total snowfall amounts.

*L 173-174: When you say "horizontally aligned", do you mean perfectly horizontally aligned, or are the particle orientations allowed to vary randomly by some maximum angle relative to horizontal (i.e., 'flutter')? My experience with discrete dipole calculations indicates the first approach can overestimate backscatter cross-sections if the radar is vertically pointing. This may not be a significant concern with a 40-degree viewing angle, though.*

The artificial particles are horizontally aligned and not fluttering. We used the Self-Similar Rayleigh Gans Approximation (SSRGA), to estimate radar backscattering cross-section. The parameterization of the SSRGA backscattering cross-section for rimed particles is described in Maherndl et al. (2023), where we evaluated the method against DDA.

*L 180: Can you provide a reference for this assertion about the dominance of riming for particle mass?*

Yes, we added two references:

> Because currently no particle classification product is available for all sites and mass-size parameter variability is rather dominated by riming than by particle shape (Maherndl et al., 2023; Mason et al., 2018), we assume a mixture of particle shapes (columns, dendrites, needles, plates, rosettes) and use the "mean" mass-size parameters, which are closest to the parameters for aggregates of plates.

*L 219-224: So, to be sure that I understand, you compared the relationship between Ze and IWC for 90-degree observations from SAIL against the corresponding relationship derived for 40-degree observations from SAIL? And the m(D) parameters by which IWCs were determined were obtained by fitting modeled reflectivities (obtained from the observed PSDs) against the observed 40-degree slant path observed Ze?*

Yes, we performed the $M$ retrieval for 90-degree observations and 40-degree observations during scans at the SAIL site, caluated IWC and compared to $Z_e$ (in the range gate closest to the in situ snowfall camera). We found a shift in $Z_e$, which we strongly assume is due to particle orientation. Yes, m(D) are obtained from fitting modeled reflectivities (from the observed PSDs) and observed reflectivities for the respective viewing angle, which is accounted for in the forward modeling.

*L 232-249: There are a few things to clarify here. First, although the relationships given in (6), (7), (8), and (9) are general, is it your intention to fit these relationships using the 40-degree Ze observations? And I'm fairly certain the 'LWP' used here is the LWP measured in the vertical direction, but it would be helpful to explicitly say so. The reason to clarify this is that at least one of the DOE microwave radiometers (the two-channel) produces both a liquid water path in the zenith direction and a liquid water path that is along the line-of-sight path of the radiometer.*

Yes, the intention is to use the 40-degree $Z_e$. We only use the lowest $Z_e$ (closest to ground) so that we can compare to the ground-based in situ data. We do not look at the (slanted) $Z_e$ column. LWP is taking vertical not slanted. We did not shift the time series because the averaging we do is sufficient, see the discussion in lines 168-171 (of the revised manuscript).

We have added the following:

> The reference IWC in $kgm^{-3}$ and SR in liquid water equivalent $mmhr^{-1}$ (Sect. 3.2) are related to the  (40° slanted) radar reflectivity factor close to ground $z_e$ in linear units $mm^6m^{-3}$, ...

and

> Therefore, we also relate the reference IWC in $kgm^{-3}$ and SR in liquid water

equivalentmmhr$^{-1}$ to $z_e$ in mm$^6$m$^{-3}$, $T$ in °C, and the vertical LWP in kgm$^{-2}$,
...

*L 273-275, Figure 4: Distinct point shapes are not apparent, particularly in the locations where the scatter plot point density is high. Would 2D histograms be more useful for panels (a) and (c)? Also, in Figure 4, there are some interesting behaviors. In panel (a), if one follows a vertical, downward trajectory of constant Ze, normalized rime mass M increases while IWC decreases. It would be interesting to see what this means in terms of size distribution changes, but that is beyond the scope of this work.*

Because we want to show $M$ colorcoded and think this information is more important than the density of data points, we decided against 2D histograms.

Thank you for the comment about size distribution changes, this would indeed be interesting to study.

*L 297-299: This explanation seems a bit simplistic. Could there not be differences versus particle habit assumptions or versus the population of PSDs used by Hogan et al. to produce their relationship? It's not apparent to me that this difference is attributable solely to the use of unrimed particles by Hogan et al.*

We agree and changed the sentence to:

> This is likely in part due to Hogan et al. (2006) using mass-size parameters for unrimed particles in  their calculations of reference IWC. Differences in particle habit assumptions and PSD observations might also play a role.

*L 314-317: At the SAIL site, it appears the LIMRAD94 was limited to a range of 2000m, or a height above ground of about 1300 m at the 40-degree observation angle. Similarly the maximum range was 2000m at Eriswil. The maximum heights for the RPG-FMCW-94-DP and JOYRAD-94 are not specified. For this WIVERN-like comparisons, did you simulate an atmospheric column? What were the depths of the columns and how did you treat 94-GHz attenuation? Also, how did you approximate the 1 km horizontal resolution using the ground-based observations? Please explain more about how the WIVERN observations are simulated from the ground observations. Finally, given the empirical nature of the retrieval, it is not clear how these uncertainties are introduced into the retrieval process and how their effects are evaluated. Please explain.*

We approximate the WIVERN observations simply by regridding and averaging the ground-based radar data to the vertical resolution planned for WIVERN and applying noise to account for the expected measurement uncertainties. The relations are then only applied to reflectiviy data in the range bin closest to ground, so that we can use the ground in situ data as a reference. We assume WIVERN data to be corrected for gaseous attenuation. We did not look at higher altitudes because we lack reference data there. In the revised text, we added:

To approximate (attenuation corrected) WIVERN $Z_e$ observations, the high resolution, ground-based data from SAIL and the additional sites are downsampled to WIVERN geometry, i.e., a vertical resolution of about 580 m and a horizontal resolution of 1 km. We consider only the lowest grid point of the regridded $Z_e$ (meaning the lowest about 580 m), which we use to compare to our reference IWC and SR. In addition, uncertainty estimates are applied to the regridded data to approximate WIVERN measurements. Uncertainties are applied in form of Gaussian noise and afterwards our relations are applied to the regridded and noisy data. $Z_e$ uncertainties are derived based on simulations (Battaglia et al., 2024); for $T$ an uncertainty of 2 K, and for LWP an uncertainty of 30 gm$^{-2}$ are assumed. 30 gm$^{-2}$ was chosen based on the maximum uncertainty of the retrievals from Ruiz-Donoso et al. (2020) and Billault-Roux, Berne (2021) (in mid- and high-latitudes) ...

*L 323-326: See my previous comment regarding how the effects of uncertainties are evaluated.*

See comment above.

*L 356-360: Wind effects ('pumping') are probably the most significant source of error for heated Pluvio guages. Did you consider filtering out cases with high wind conditions to see if the validation results improved?*

The in situ data is also less reliable under high wind conditions (Maahn et al., 2024). By applying the VISSS blowing snow filter (see VISSS processing library https://github.com/maahn/VISSSlib), high wind speed events are mostly filtered out.

*L 380-383: See my earlier comment. At face value, increased riming of particles should increase IWC. But in this case, you are describing the changes given a constant Ze. So, what physical processes are happening if Ze is constant but riming increases and IWC decreases? It would be helpful to most readers to provide some insight here.*

We were not talking about a physical process keeping $Z_e$ constant here, but rather about a space of possibilities. At a given $Z_e$ one particle population made up of heavily rimed particles has a lower IWC than another particle population made up of unrimed particles. This is due to enhanced scattering of the heavily rimed particles. We have changed to:

> For a  given $Z_e$,  ice particle populations have a lower IWC the more heavily rimed they are due to the enhanced scattering of rimed particles (Fig. 4).

*L 391-395: Se my earlier comment. More details are needed in the discussion section regarding how these WIVERN-like measurements were produced and how observational errors were propagated into the empirical retrieval.*

13

See comment above.

**Minor language corrections**

*L 40: Should be 'particle size distributions'.*

*L 58: Should be 'signficantly reducing the uncertainty'.*

*L 295: Should be 'site'.*

*L 298: Should be 'their'.*

*L 357: Should be 'HYY is operated as'.*

Thank you, we corrected the language errors.

---

## Author Response (AR2)

**Riming-dependent Snowfall Rate and Ice Water Content Retrievals for W-band cloud radar**

N. Maherndl, A. Battaglia, A. Kötsche, and M. Maahn

April 26, 2025

*Original Referee comments are in italic*

manuscript text is indented, with added text underlined and removed text crossed out.

We would like to thank the reviewer for their helpful comments. We revised the manuscript and responded to all of the reviewer's comments.

**Reviewer I**

*Thank you for the answers to my comments.*

*"It must be noted that our reference IWC and SR data are not fully independent of Ze because we derive the particle mass from the retrieved normalized rime mass M . This is a necessary limitation because IWC and SR cannot be inferred from the available in situ measurements alone. For SR, we evaluate our approach with completely independent SR gauge measure"*

*I am fully aware that this is a necessary limitation. What I think would be good is to give the reader some discussion of the implications of this limitation.*

*For example, in the review answers, you mention: "If Ze would have a positive bias, then M would have a positive bias as well, resulting in a positive bias of IWC" That means in Fig. 4a, the data points would move towards higher Ze and IWC at the same time. Therefore, e.g. the slope of the fit in Fig. 4b would be unaffected?*

Yes, we presume that the slope in Fig. 4b would not be affected by a positive bias in Ze. However, the exact relation (i.e. a positive bias of x dB results in a factor y higher IWC) depends on the PSD and can therefore not be given universally.

In the manuscript, we added:

> As discussed in Sect. 3.2, our reference IWC and SR are not fully independent of $Z_e$ due to the dependence on $M$. If $Z_e$ would have a positive bias, then $M$ would have a positive bias as well, resulting in a positive bias of IWC or SR. The slope of the fits in Fig. 4b,d would therefore likely not be affected by a bias in $Z_e$.

*Or, as another example: For a given Ze (e.g. 5 dBZ), the measurements in Fig. 4a show a nice correlation between M and IWC: all the yellow points lie at the lower edge of the point cloud, all the black points at the upper edge. But this pattern is a direct consequence of the underlying retrievals and the fact that M and IWC are derived from the same parameters. It is not something the data reveals to us, if I am correct.*

Not only M, but also the PSD impacts Ze and IWC. Assuming a fixed PSD shape, the pattern is indeed a consequence of the underlying retrievals: for a given IWC, heavily rimed particles result in higher Ze than unrimed particles when using the riming-dependent scattering parameterization from Maherndl et al. (2023). However, variability in PSD shape (e.g., a heavily rimed particle population with many small, but few large particles vs. an unrimed particle population with few small and many large particles) results in variability in the pattern. We therefore do not fully agree with the statement that the pattern says nothing about the data.

In the manuscript, we added:

> PSD shape also affects the pattern in Fig. 4a,c. Assuming a fixed PSD, the spread in IWC-$Z_e$ space due to riming is a direct result of the underlying riming-dependent paramterization (Maherndl et al., 2023). However, variability in (observed) PSDs results in variability in the pattern.

*You also mention: "We currently don't have size resolved measurements of particle mass" As an outlook: If you would have such measurements, could this lead to different results or improvements in the relations? If this could be measured in the near or far future, would it be worth to repeat the study?*

Size-resolved measurements of particle mass would lead to improvements by reducing uncertainties in the mass-size assumptions we make. Additionally, a completely independent in situ IWC could be derived, which would be a better reference IWC.

In the outlook, we added:

Further observational studies focusing on particle mass and scattering behavior are needed to investigate these assumptions. Uncertainties due to the mass-size assumptions could be reduced if size-resolved particle mass observations were available in the future. This way, completely independent reference IWC and SR could be derived with which the study should be repeated.

*I think openly discussing such points would in no way reduce the importance or validity of the results of this publication, but give the reader a more holistic and nuanced view on the topic.*

Thank you for the comments, the point is well taken. We have added further discussion in regards to the implications of the study limitations (see comments above).

**Technical Notes**

*For a very high density of points like in Fig. 4a, scatterplots are not the optimal choice. With so many overlapping points, the final color of the plot is basically determined by the order in which the points are plotted, as well as the rendering settings of the plotting framework. In this case, a 2D histogram, where the mean or median M value is calculated for all points in the same bin, would be more appropriate.*

We agree that scatterplots are not optimal for a high density of points. However, a 2D histogram or a color mesh plot is also not ideal for the information we want to show. In Fig. 4a,c we aim to show two things: 1. the density of data points and 2. the dependency of IWC-Z and SR-Z on $M$. If we only want to show 1., a 2D histogram would definitely be better. If we only want to show 2., a binned plot like you suggest with the median or mean $M$ colored would be best. We think to show both 1. and 2., a scatterplot with semi-transparent, colored markers is a good compromise. We think that increasing the number of plots or panels does not add much value in terms of additional information and makes the plot more complicated and harder to read at first glance. We therefore have not changed Fig. 4 (yet). However, given the subjectivity of the matter, we are not opposed to changing the plot. If requested by the editor, we will provide an updated plot with an additional panel to display 1. and 2. separately as 2D histograms (1. showing the data density and 2. showing the average $M$).